# Neural effects of propofol-induced unconsciousness and its reversal using thalamic stimulation

André M Bastos[1‡§], Jacob A Donoghue[1‡], Scott L Brincat[1], Meredith Mahnke[1], Jorge Yanar[1], Josefina Correa[1], Ayan S Waite[1], Mikael Lundqvist[1], Jefferson Roy[1], Emery N Brown[1,2,3†], Earl K Miller[1†*]

[1]The Picower Institute for Learning and Memory and Department of Brain and Cognitive Sciences, Massachusetts Institute of Technology, Cambridge, United States; [2]The Department of Anesthesia, Critical Care and Pain Medicine, Massachusetts General Hospital/Harvard Medical School, Boston, United States; [3]The Institute for Medical Engineering and Science, Massachusetts Institute of Technology, Cambridge, United States

**Abstract** The specific circuit mechanisms through which anesthetics induce unconsciousness have not been completely characterized. We recorded neural activity from the frontal, parietal, and temporal cortices and thalamus while maintaining unconsciousness in non-human primates (NHPs) with the anesthetic propofol. Unconsciousness was marked by slow frequency (~1 Hz) oscillations in local field potentials, entrainment of local spiking to Up states alternating with Down states of little or no spiking activity, and decreased coherence in frequencies above 4 Hz. Thalamic stimulation 'awakened' anesthetized NHPs and reversed the electrophysiologic features of unconsciousness. Unconsciousness is linked to cortical and thalamic slow frequency synchrony coupled with decreased spiking, and loss of higher-frequency dynamics. This may disrupt cortical communication/integration.

**\*For correspondence:**
ekmiller@mit.edu

[†]These authors also contributed equally to this work
[‡]These authors also contributed equally to this work

**Present address:** [§]Department of Psychology and Vanderbilt Brain Institute, Vanderbilt University, Nashville, United States

**Competing interests:** The authors declare that no competing interests exist.

## Introduction

Propofol – the most widely used anesthetic – acts by enhancing GABAergic inhibition throughout the brain and central nervous system (*Bai et al., 1999*; *Hapfelmeier et al., 2001*; *Hemmings et al., 2019*; *Hemmings et al., 2005*). In humans, propofol produces dose-dependent changes in arousal level that are associated with spatiotemporal neurophysiological signatures across the cortex. At low doses, propofol produces a sedative state associated with beta oscillations (13–25 Hz) (*McCarthy et al., 2008*; *Purdon et al., 2015*). At higher doses that maintain unconsciousness for surgery, slow-delta oscillations (0.1–4 Hz) (*Lewis et al., 2012*; *McCarley, 2007*) appear across the entire scalp and are decoupled between cortical regions (*Lewis et al., 2012*). Concomitantly, coherent alpha oscillations (8–12 Hz) concentrate across the frontal area of the scalp, a process known as anteriorization (*Cimenser et al., 2011*; *Purdon et al., 2013*). In profound states of propofol-induced unconsciousness, the phase of the slow-delta oscillations strongly modulates the amplitude of the alpha oscillations (*Purdon et al., 2013*).

The broad range of dynamics observed in response to propofol administration suggests that propofol-induced unconsciousness is a multifactorial process. One factor is that unconsciousness is caused by the phase-locking of neuronal spiking with the slow-delta oscillation. This greatly reduces cortical spiking and limits cortical activity to brief Up-states of spiking followed by longer duration Down-states of little or no spiking (*Lewis et al., 2012*). Another factor is that the slow-delta oscillations 'fragment' the cortex (*Lewis et al., 2012*). Local spiking becomes limited to the narrow window

of slow-delta oscillation phases, which being decoupled across cortical areas, impede long-range cortical communication. A third factor, supported by modeling and experimental studies, suggests that the coherent frontal alpha oscillations represent hypersynchronous communication between the thalamus and prefrontal cortex (*Ching et al., 2010*; *Flores et al., 2017*; *Palva and Palva, 2007*). A fourth factor links unconsciousness to loss of frontal-parietal connectivity (*Lee et al., 2013*). Finally, the neuroanatomy of the brainstem suggests that direct action of propofol at GABAergic inhibitory synapses onto the arousal center nuclei is also an important contributor to propofol-induced unconsciousness (*Brown et al., 2010*).

Details of propofol-induced unconsciousness remain to be clarified because its dynamics have been investigated mostly using extracranial measures such as electroencephalography (EEG) with limited spatial specificity and fMRI with limited temporal specificity (but see *Redinbaugh et al., 2020*). Microelectrode recordings in patients have highly precise temporal resolution but limited spatial coverage (*Lewis et al., 2012*). Studies conducted simultaneously at high spatial and temporal resolution are needed. With one exception (*Flores et al., 2017*), previous studies have not included the thalamus, a critical nexus that regulates cortical activity (*Saalmann and Kastner, 2015*; *Sherman, 2016*). The thalamus is highly interconnected with the cortex and receives important inputs from the brainstem arousal centers (*Jones, 2007*). Furthermore, both thalamo-cortical and cortico-thalamic connectivity are highly layer specific, giving rise to specific hypotheses that propose either deep (layers 5/6) or superficial (layer 2/3) may be more specifically linked to loss of consciousness (*Aru et al., 2019*; *Dehaene and Changeux, 2011*). *Dehaene and Changeux, 2011* and others link consciousness more to frontal vs. posterior cortex (*Boly et al., 2017*). Central thalamic stimulation can result in a partial restoration of consciousness (*Schiff et al., 2007*). Some theories suggest consciousness is formed when oscillations in thalamo-cortical loops integrate cortical information (*Llinás, 2003*). There is mounting evidence that the thalamus regulates cortical communication (especially top-down) via oscillatory dynamics (*Halassa and Sherman, 2019*; *Saalmann and Kastner, 2015*).

To study the effects of propofol on cortex and thalamus, we administered propofol to four macaque monkeys via vascular access ports or implanted catheters using computer-controlled infusions. We continuously recorded neural activity as the animals transitioned from the pre-propofol awake state, through loss of consciousness (LOC), to unconsciousness and recovery of consciousness (ROC) (*Figure 1A*).

## Results

### Experimental design, recordings, and physiological responses to propofol

We started by administering a high-infusion rate of propofol (280–580 mcg/kg/min, adjusted per individual animal) for 20 min to induce LOC (defined as the moment the eyes closed and remained closed for the remainder of the infusion). LOC was associated with simultaneous changes in several physiological variables. We assessed the significance of the change in each physiological variable as the posterior probability that its values were greater during the Awake state than they were 10–60 min after initiating the propofol infusion (*Smith et al., 2005*). We considered a change to be significant if the posterior probability was greater than 0.99.

Relative to the Awake state, we observed a significant decrease in muscle tone (from 4.5 to 75 min, *Figure 1C*), cessation of airpuff-evoked eyeblinks (from 1 to 75 min, *Figure 1D*), a decline in blood oxygenation (from 4.5 to 34 min, *Figure 1E*), and a decrease in heart rate (from 4.5 to 75 min, *Figure 1F*). All times were measured relative to the start of the propofol infusion. After LOC, we decreased the infusion to an animal-specific maintenance rate (140–230 mcg/kg/min) for an additional 30–45 min. Once the infusion was ceased, ROC occurred in ~7 min. ROC was behaviorally defined as the moment the eyes opened and remained open continuously. We will refer to the pre-propofol awake state as the Awake state and the period from LOC to ROC as the Unconscious state.

We performed the neurophysiological recordings in three sets of experiments. In the first set of experiments, we recorded neuronal spiking activity and local field potentials (LFPs) from a series of chronically implanted 64-channel 'Utah' arrays in prefrontal (8A and PFC), posterior parietal area 7A/

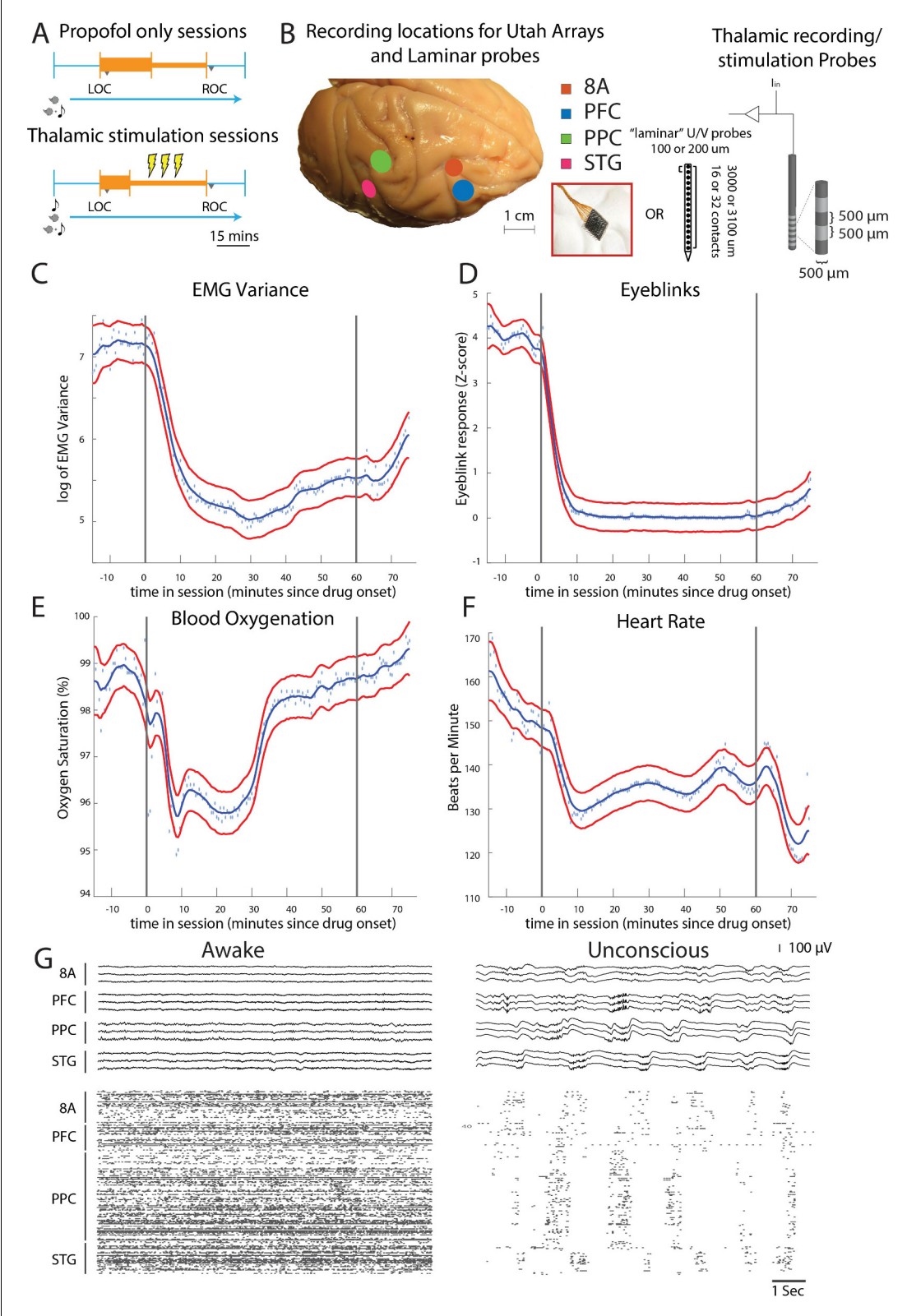

**Figure 1.** Propofol anesthesia paradigm and physiological indices of LOC. (**A**) Session paradigm. Two sets of sessions were performed. For propofol-only sessions (upper subplot), there was an initial 30 min infusion (fast rate, thick orange bar) covering Awake (pre-LOC) and Unconscious states, before switching to a halved rate propofol infusion for the maintenance phase of experiment (narrow orange bar). For the thalamic stimulation sessions (N = 22), the initial infusion was for 20 min, followed by a halved rate propofol infusion for the rest of the session. Periodically, 28.5 s trials with electrical

*Figure 1 continued on next page*

stimulation in the thalamus (yellow bolts) occur during lower-dosed maintenance phase of propofol infusion. LOC: loss of consciousness, ROC: recovery of consciousness. (B) (left) Cortical recording locations of each 64-channel chronic recording arrays or 16/32 channel acutely inserted laminar probe. PFC: ventrolateral prefrontal cortex; 8A: caudal lateral PFC; PPC: posterior parietal cortex area 7A/7B; STG: superior temporal gyrus; (right) 'Laminar' probes and thalamic electrical stimulation/LFP (Local Field Potential) recording leads. (C-F) Physiological measurements characterizing the Awake state relative to propofol administration (starting at time zero). Blue dots indicate individual time points with measurements averaged across sessions. Blue curve is a smoothed estimate and the red curves are the approximate 95% confidence intervals (see 'Methods'). (G) (upper panel) Example LFP traces from all cortical Utah arrays during the Awake (left) and Unconscious states with clear slow-frequency waves (right). (lower panel) Example spike raster over 10 s of data. Spike times are indicated with dots.

B (PPC), and auditory/temporal (STG) cortex. These initial sessions (10 in monkey 1, 11 in monkey 2) served to establish the neurophysiological properties defining the Awake and Unconscious states. In the second set of experiments, we implanted multiple-contact stimulating electrodes (two in each hemisphere of the same monkeys) in frontal thalamic nuclei (intralaminar nuclei, ILN, and mediodorsal nuclei, MD, with a few sites in neighboring thalamic nucleus VPL) to record from and electrically stimulate the thalamus and cortex during the Unconscious state (a total of 22 additional sessions: 11 in monkey 1, 11 in monkey 2). In the third set of experiments conducted in two additional monkeys (a total of eight sessions: two in monkey 3 and six in monkey 4), we performed acute laminar recordings from the same areas using multi-contact arrays positioned approximately perpendicular to cortex (as in *Bastos et al., 2018*, see 'Methods'). Data from monkeys 3 and 4 were used only for the analyses involving cortical layers (Figure 5). Data from monkeys 1 and 2 were used for all other analyses. All main results are presented as averages across monkeys 3 and 4 (Figure 5) or 1 and 2 (all other figures).

An example session of cortical recordings during the Awake and Unconscious states (*Figure 1G*) showed a profound reduction in spiking, an increase in both slow-frequency (SF, 0.1-2Hz) local field potential (LFP) amplitude (*Flores et al., 2017*) and slow-frequency modulation of spiking activity.

## Changes in LFP power with the Unconscious state

We first characterized the changes in LFP power during the Awake state (*Figure 2*) to every time point after drug administration until 10 min post-ROC (non-parametric cluster-based randomizations, corrected for multiple comparisons; all effects $p < 0.01$; *Maris and Oostenveld, 2007*). We compared LFP power increases/decreases from the Awake state, time locked either to LOC or ROC. First, approximately 3–7 min before LOC, there was an increase in beta power in all areas (~15–30 Hz). In frontal areas PFC and 8A, this power change shifted to a lower frequency (~14–20 Hz) during the maintenance phase (*Figure 2A and B*, left subplots). Power in this frequency range remained higher than Awake throughout the Unconscious state. By contrast, in posterior areas PPC and STG after LOC (dotted black lines at time point 0 in *Figure 2*, left subplots) beta power was reduced relative to Awake (*Figure 2C and D*, left subplots). In all areas, shortly before LOC there was a sustained reduction in gamma power (>35 Hz).

All areas also showed a significant increase in slow-frequency power around LOC (*Figure 2*, non-parametric cluster-based randomizations, corrected for multiple comparisons; all effects $p < 0.01$). We quantified the effect size by comparing power in the slow frequency band (0.1–2 Hz) while Awake to power while Unconscious (*Figure 2—figure supplement 1A*, blue and orange bars, respectively). We measured the effect size in dB units ($10*\log_{10}$[power Awake/power Unconscious]) across areas. The order of areas from strongest to weakest change was organized from posterior to anterior. The strongest increase in SF power was seen in STG, then PPC/PFC, then 8A (*Figure 2—figure supplement 1B*, significant differences in effect sizes were observed between STG and PFC and STG and 8A, $p < 0.01$, non-parametric randomization test).

The thalamus showed similar power modulation results as cortex. There was an increase in slow frequency oscillations locked to LOC and a decrease in this same band locked to ROC (*Figure 2—figure supplement 2*, $p < 0.01$, non-parametric randomization test). In the beta frequency range, thalamic power modulation looked like a mixture between anterior and posterior cortex. There was an initial increase in beta power compared to Awake starting at LOC. This was followed by a decrease relative to Awake from ~5–14 min post-LOC. Subsequently (from 15 min post-LOC to ROC), there

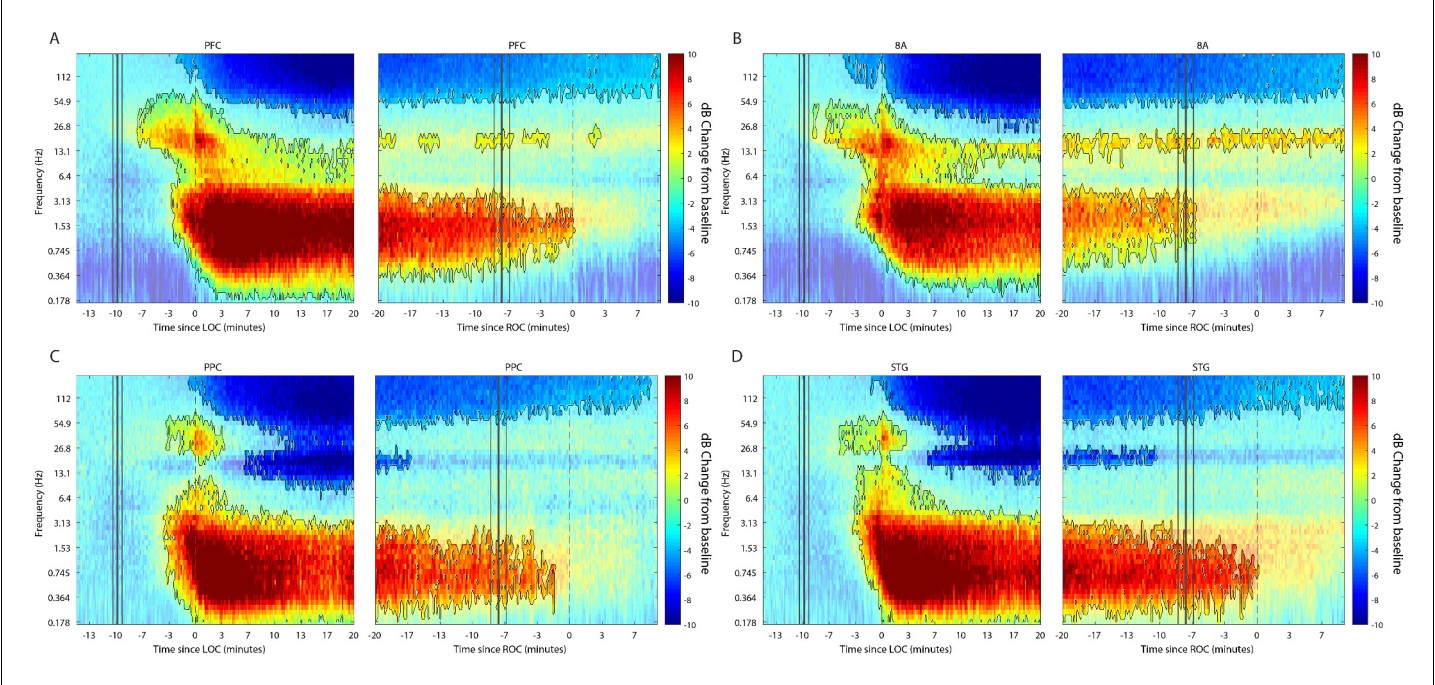

**Figure 2.** Changes in cortical power during the Unconscious state. (A-D) dB change in power for each area is shown with respect to Awake. Increases in power are shown in red, decreases in blue. Significant modulation of power is shown in opaque colors and outlined in black (p<0.01, non-parametric cluster-based randomizations, corrected for multiple comparisons). Left subpanels are time locked to loss of consciousness (LOC), defined behaviorally as the moment the eyes closed and remained closed. Start of drug infusion is shown as a vertical black bar at −10 min from LOC, ± 1 SEM over sessions. Right subpanels are time locked to return of consciousness (ROC), defined behaviorally as the moment the eyes first opened. Cessation of drug infusion is shown as a vertical black bar at −8 min from ROC, ± SEM over sessions.

The online version of this article includes the following figure supplement(s) for figure 2:

**Figure supplement 1.** Effect sizes for change in slow frequency power during the Unconscious state.

**Figure supplement 2.** Changes in thalamic power during the Unconscious state dB change in power for thalamic sites is shown with respect to the Awake state.

was neither a significant increase nor decrease in beta power relative to Awake (*Figure 2—figure supplement 2*, p>0.01 non-parametric randomization test).

## Anteriorization of alpha-beta power and differences in slow frequencies across areas during the Unconscious state

'Alpha anteriorization' is the decrease in alpha-beta power in posterior cortex and its increase in frontal cortex during the Unconscious state (*Cimenser et al., 2011*; *Tinker et al., 1977*). We plotted the effect size (10*log$_{10}$[power Awake/power Unconscious]) as a function of frequency (*Figure 3A*). This revealed differences between posterior and anterior areas in the alpha-low beta range. In particular, area 8A showed a clear peak at 15 Hz during unconsciousness (*Figure 3A*, red line). At the same frequency and above, posterior brain regions were strongly depressed in power (*Figure 3A*, green and magenta lines). We directly compared the absolute power in all areas between the Awake and the Unconscious states. We found that in the Awake state, alpha-beta (8–25 Hz) power in posterior areas, PPC and STG, was greater than in anterior areas, PFC and 8A (*Figure 3B*, sign test for anterior vs. posterior alpha-beta power across sessions, p<1E-3 for all comparisons). During the Unconscious state this relationship flipped. Alpha-beta power in frontal areas PFC and 8A was greater than in the posterior areas (*Figure 3C*, sign test for anterior vs. posterior alpha-beta power across sessions, p<1E-5 for all comparisons). These observations suggest the presence of anteriorization.

This analysis revealed differences in which frequencies the slow frequency band changed the most between Awake and Unconscious states. In anterior areas, PFC and 8A, the peak change was

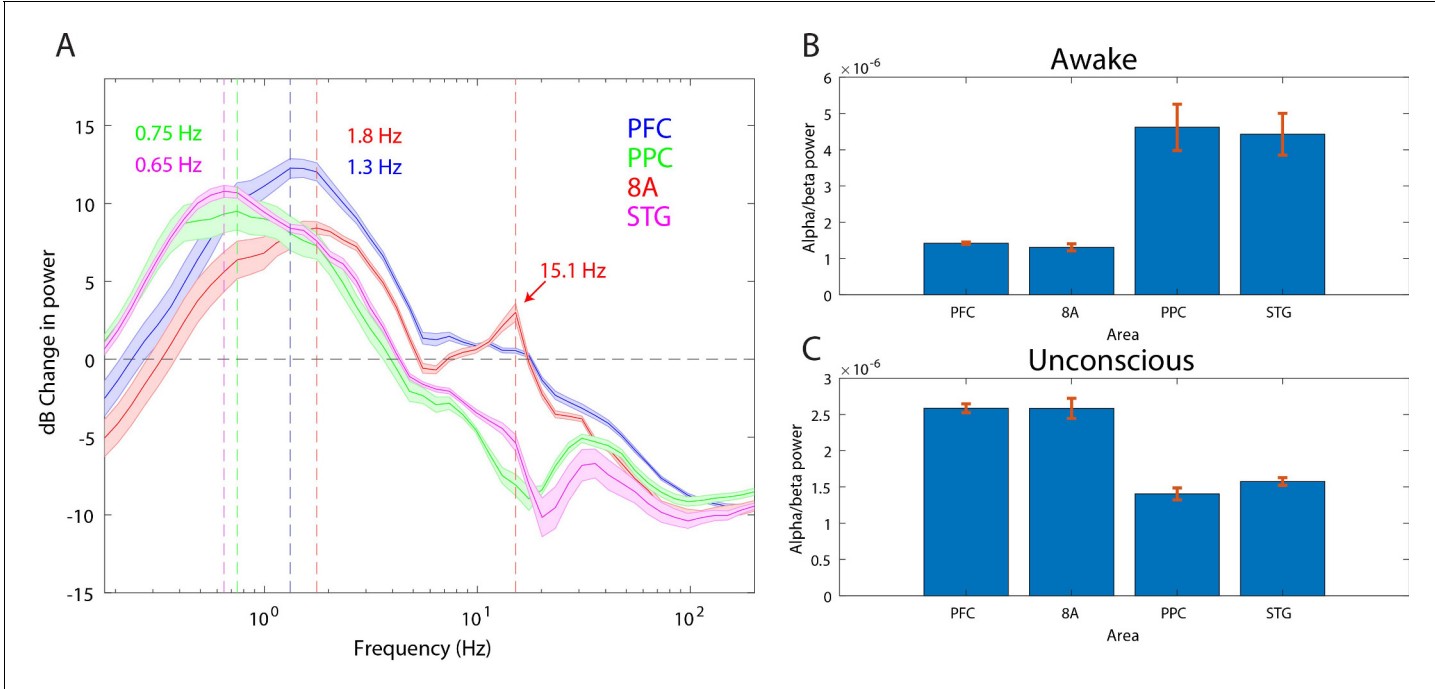

**Figure 3.** Power asymmetries between areas. (**A**) dB change in power of Unconscious vs. Awake. Positive values indicate power Unconscious > Awake. For each area, the mean power difference is shown 1 SEM. Peak power modulation in the slow frequency range is highlighted for each area, and a secondary peak in the beta range (15.1 Hz) is present in area 8A. (**B**) Power in the beta frequency range (8–25 Hz) during the Awake state. (**C**) Same as B, but during the Unconscious state. Mean ±/- 1 SEM. PFC: ventrolateral prefrontal cortex; 8A: caudal lateral PFC; PPC: posterior parietal cortex area 7A/7B; STG: superior temporal gyrus.

in higher frequencies (*Figure 3A*, 1.3 and 1.8 Hz, respectively) than posterior areas, PPC and STG (*Figure 3A*, 0.75 and 0.65 Hz, respectively, sign test for frequency difference, all anterior vs. posterior comparisons, p<1E-5).

## LFP power changes from unconscious to ROC

ROC occurred ~8 min after the cessation of the propofol infusions. In all areas, at ROC, power in the slow-frequency range was no longer significantly different from the Awake state (*Figure 2A–D*, right subpanels). In contrast, significant reductions in gamma power were still present in PFC, 8A, and STG even 10 min post-ROC. Beta power changes from Awake were also not reliable indicators of ROC. In frontal area 8A, beta power remained elevated above Awake levels 10 min post-ROC (*Figure 2B*, right subpanel). In posterior areas, beta power reductions from Awake were no longer significant at 10–17 min pre-ROC.

## Differences in spiking between Awake vs. Unconscious

We next examined differences in spike rates between the Awake and Unconscious states (*Figure 4A*). We eliminated periods with airpuffs or auditory tones in order to not induce differences across areas due to differences in sensory processing. In the Awake state, average spike rates were between 6 and 8 spikes per second across all areas. During the Unconscious state, average spike rates across areas fell to 0.2–0.5 spike/s during the initial infusion (first 30 min of propofol). They increased to 1–2 spikes/s during the maintenance infusion. After cessation of propofol, spike rates gradually approached the levels seen in the Awake state (*Figure 4B*).

At LOC, spike rates were approximately half of their Awake rates (*Figure 4A*, mean firing rates at LOC: 4.0, 3.2, 3.4, 3.8 spikes/s for PFC, 8A, PPC, STG, respectively). Spike rates continued to fall and then stabilized ~15 min post-LOC to an average of 0.25 spikes/s. During the maintenance infusion, average spike rates across all areas increased to 1–1.7 spikes/s.

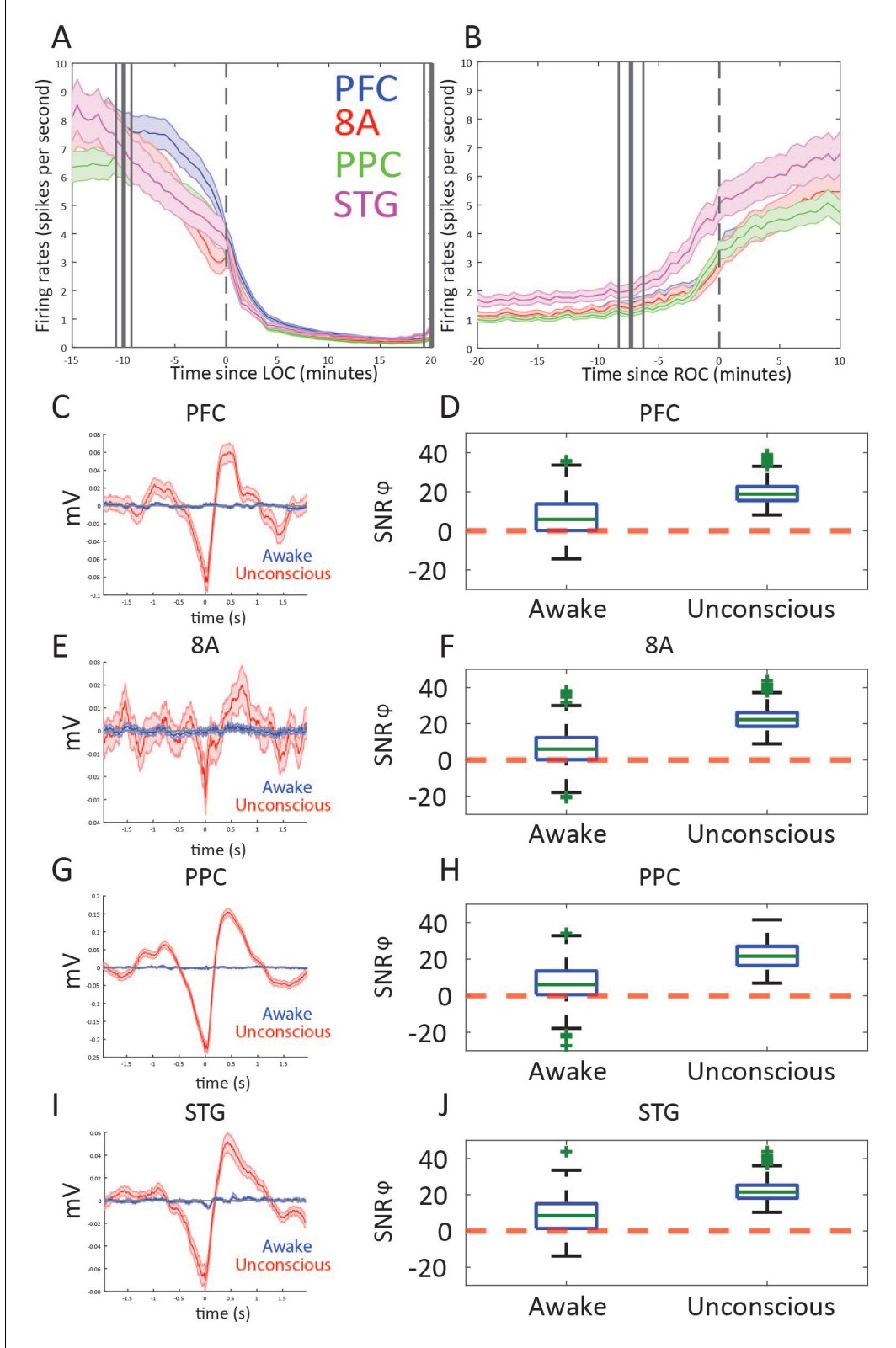

**Figure 4.** Changes in mean firing rate and spike-phase modulation during Awake vs. Unconscious states. (**A**) Spike rate for all recorded areas averaged across all propofol-only recording sessions locked to loss of consciousness (LOC). (**B**) Same as **A**, but for recovery of consciousness (ROC). Mean and 99 percent confidence interval. (C/E/G/I) The spike-triggered average for all well-isolated units in a given area with respect to that area's unfiltered LFP.

*Figure 4 continued on next page*

*Figure 4 continued*

Red is the Unconscious state, blue is the Awake state. (D/F/H/J) Boxplots of SNRΦ values in decibels (dB) (reflecting SF modulation of spiking, see 'Methods') for all units during the Awake and Unconscious states. PFC: prefrontal cortex; PPC: posterior parietal cortex 7A/7B; 8A: caudal lateral PFC; STG: superior temporal gyrus.

The online version of this article includes the following figure supplement(s) for figure 4:

**Figure supplement 1.** Effect sizes for change in spiking during the Unconscious state.

At ROC, spiking activity increased to 3.3, 3.0, 3.4, and 5.0 spikes/s (for PFC, 8A, PPC, STG, respectively), approximately the same spike rate as at LOC (*Figure 4B*). The exception was area STG, which recovered faster than the other areas. Neurons in STG had an average of 5.0 spikes/s at ROC, significantly greater than the other areas (p<0.01, non-parametric randomization test for all area comparisons to STG).

We quantified the effect size by comparing spike rates during the Awake to the Unconscious state (*Figure 4—figure supplement 1A*). We measured the effect size in dB units ($10 \cdot \log_{10}$[firing rate Awake/spike rate Unconscious]) across areas. The order of areas from highest to lowest spike rate change showed an anterior to posterior organization. The strongest effects were in PFC/8A, then PPC, then STG (*Figure 4—figure supplement 1B*, all comparisons, p<0.01, non-parametric randomization test).

During the Unconscious state, spike timing was phase-coupled to the slow frequencies. This appeared as Up and Down states of high vs. minimal to no spiking, respectively (e.g., *Figure 1G*). Spike-triggered averages of the LFP signal indicated that spikes entrained to the depolarized phases (troughs) of slow frequency oscillations (*Figure 4C/E/G/I*) in all areas. We further estimated the contribution of phase to predicting spiking using a generalized linear model (GLM) framework (see Methods). *Figure 4D/F/H/J* shows boxplots of the contribution of phase to predicting spiking in each area during the Awake and Unconscious states measured in dB as the signal-to-noise ratio of slow-frequency phase. We defined an increase in phase-modulation during the Unconscious state relative to the Awake state as a log-fold change in the SNRΦ (termed ΔSNRΦ) greater than zero. We computed the posterior probability of an increased phase modulation (Unconscious > Awake) across the population of neurons recorded in each area using the beta-binomial model (*DeGroot and Schervish, 2012*). The posterior probability of an increase in phase modulation was 0.99 for PFC, PPC, and 8A, and 0.98 for STG.

## Laminar changes in spiking and LFP power during the Unconscious state

In two additional animals (monkeys 3 and 4) we used multi-laminar probes to examine differences between cell layers (*Figure 5*). We pooled data from PFC, 8A, PPC, and STG at the slow frequency and gamma bands because these frequencies behaved consistently across areas. Laminar position zero was based on the relative power profile of gamma vs. beta (see Methods). This was previously shown to map onto the location of layer 4 (*Bastos et al., 2018*). As before, we calculated the dB change from the Awake to the Unconscious state. The laminar profile of change in spiking, gamma, and slow frequencies are shown in *Figure 5A/C/E*. Spiking and gamma reduction were more pronounced in superficial layers compared to deep layers (*Figure 5B and D*, non-parametric randomization test comparing the effect size of superficial vs. deep, p<1E-7 for spiking, p<1E-7 for gamma; *Manly, 2018*). The enhancement of slow frequency power was higher in deep compared to superficial layers (*Figure 5F*, non-parametric randomization test comparing the effect size of superficial vs. deep, p<1E-5 for SF).

## Changes in cortico-cortical and thalamo-cortical phase synchronization during the Unconscious state

We next analyzed whether slow-frequency LFP oscillations were also phase synchronized. We used a sliding window approach to quantify the pairwise phase consistency (*Vinck et al., 2010*; see 'Methods') continuously from 15 min pre-LOC to 10 min post-ROC. There were strong increases in slow frequency cortico-cortical phase synchronization between all cortical areas (p<0.01, cluster-based

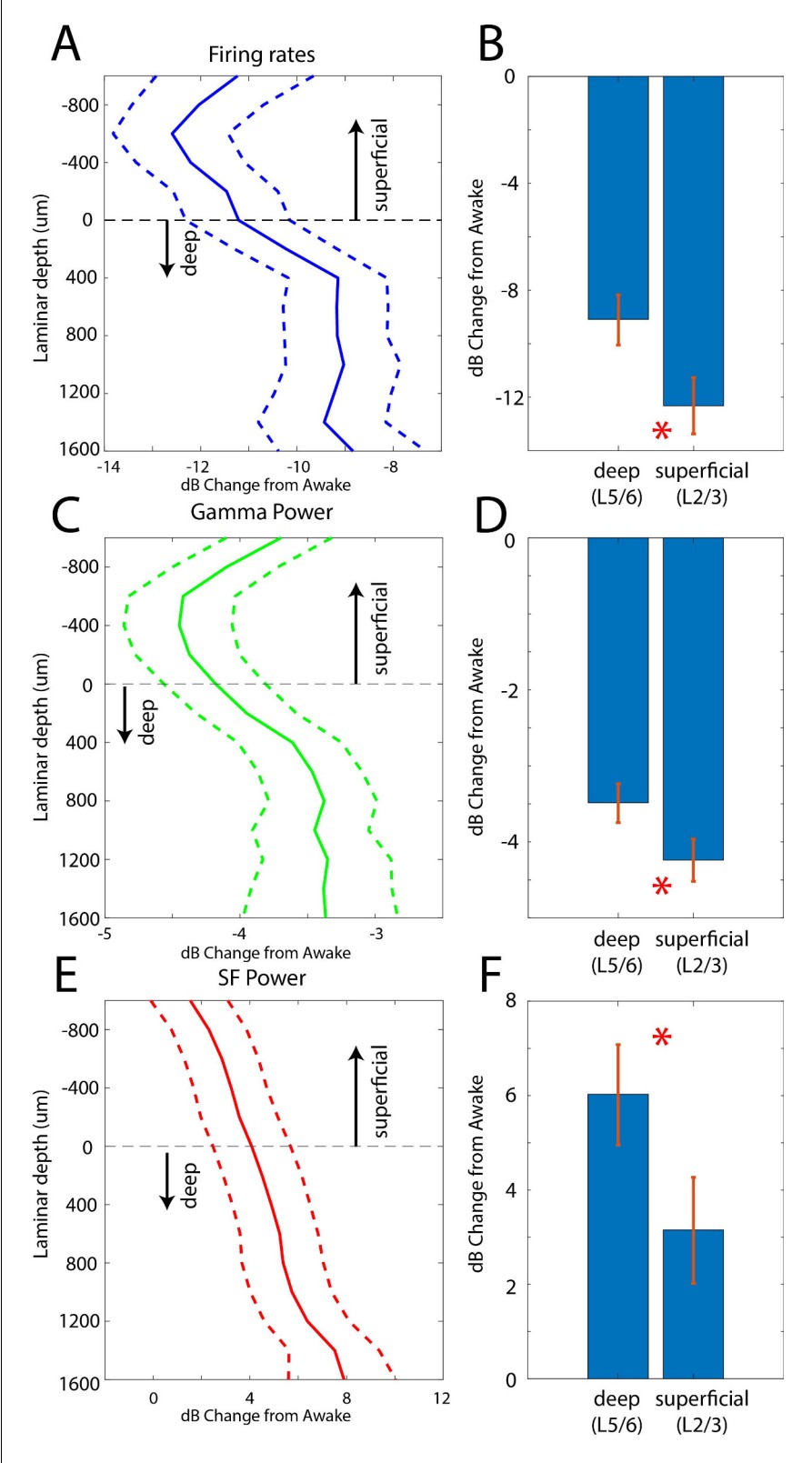

**Figure 5.** Laminar changes in spiking, gamma, and SF power during Awake vs. Unconscious states. (**A**) Firing rate change from Awake to Unconscious states as a function of cortical layer. Negative values indicate less spiking during the Unconscious state. Layer 0 is the approximate location of cortical layer 4 (see 'Methods'). The horizontal dotted line at zero separates superficial layers 2/3 from deep layers 5/6. Mean and 99 percent confidence interval of the effect size across all neurons per indicated depth. (**B**) Mean and 99 percentconfidence interval of the effect size across all superficial (N = 287)

*Figure 5 continued on next page*

*Figure 5 continued*

and deep (N = 337) neurons. (**C, D**) Same as **A, B**, but for LFP power at gamma (100–200 Hz). (**E, F**) Same as **A, B**, but for SF power (0.2–1.1 Hz). Positive values indicate more power during the Unconscious state. N = 330 LFPs for superficial layers, N = 393 LFPs for deep layers. Mean and 99 confidence interval across all available LFPs in each layer.

non-parametric randomization test, *Figure 6*) during the Unconscious state. In a subset of area pairs (*Figure 6B/E/F*), there were reductions in cortico-cortical theta and beta phase synchronization, although the effect size at these frequencies was smaller than at slow frequencies (*Figure 6—figure supplement 1*). Unique to phase synchronization between frontal areas PFC and 8A, we observed a sustained increase in alpha (7–15 Hz) frequency range phase synchronization during the Unconscious state (p<0.01, cluster-based non-parametric randomization test, *Figure 6A*).

Recent evidence suggests that the thalamus helps foster synchrony between cortical areas (*Saalmann and Kastner, 2015*). During the Unconscious state, all cortical areas show increased slow frequency phase synchronization with the thalamus (p<0.01, cluster-based non-parametric randomization test, *Figure 7*). There were also reductions in synchrony while Unconscious in the theta/beta

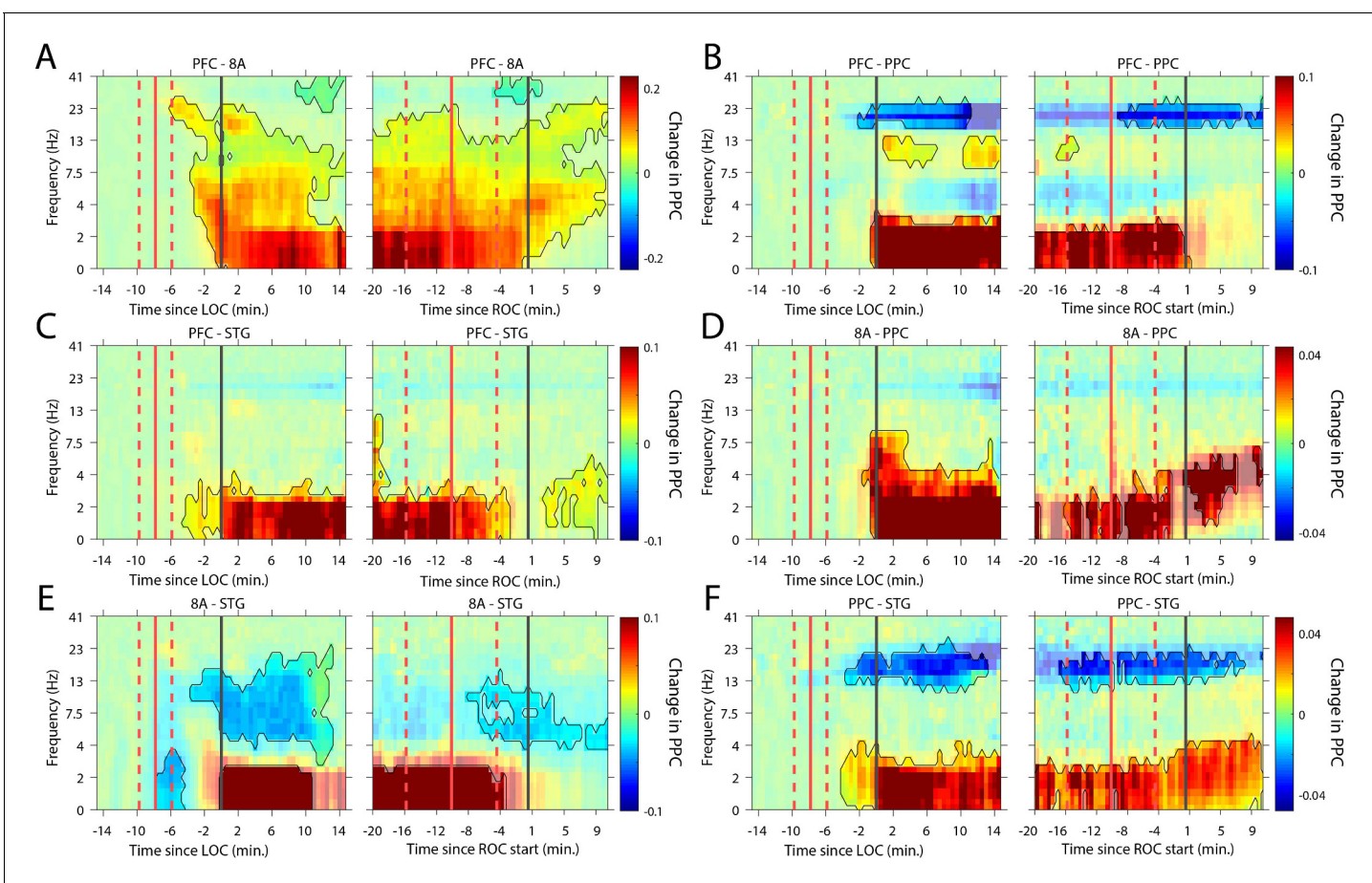

**Figure 6.** Changes in cortico-cortical LFP phase synchronization during Awake vs. Unconscious states. (**A-F**) (left subpanels) Change in the pairwise phase consistency (PPC) for all time points relative to loss of consciousness (LOC) compared to the Awake state (−15 to −10 min pre-LOC). The red vertical lines indicate the average ±/- 1 standard deviation time of propofol onset. The black vertical line indicates time zero (LOC). Significant increases or decreases (p<0.01, corrected for multiple comparisons) from Awake are opaque colors and are highlighted. (right subpanels) Same as left subpanels but for recovery of consciousness (ROC). The red vertical lines indicate the average ±/- standard deviation time of propofol offset. The black vertical line indicates time zero (ROC). Significant increases or decreases from Awake are opaque colors and are highlighted.

The online version of this article includes the following figure supplement(s) for figure 6:

**Figure supplement 1.** Effect sizes for change in cortico-cortical LFP phase synchronization during Awake vs. Unconscious states (left panels).

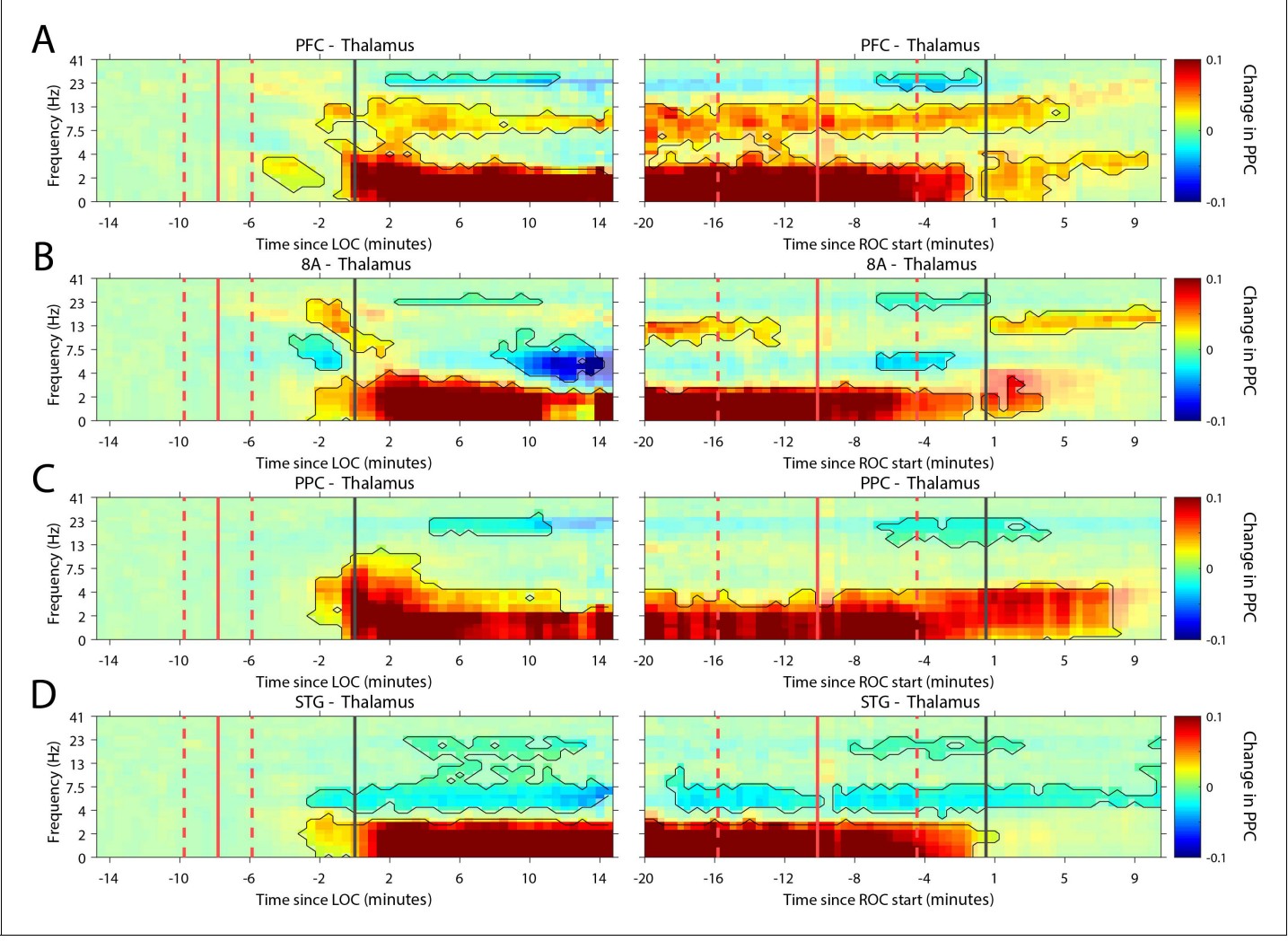

**Figure 7.** Changes in thalamo-cortical phase synchronization during Awake vs. Unconscious states. (**A-D**) (Left subpanels) Change in the pairwise phase consistency (PPC) for all time points relative to loss of consciousness (LOC) compared to Awake (−15 to −10 min pre-LOC). The red vertical lines indicate the average ± 1 standard deviation time of propofol onset. The black vertical line indicates time zero (LOC). Significant increases or decreases (p<0.01, corrected for multiple comparisons) from Awake are marked with opaque colors and are highlighted. (Right subpanels) Same as left subpanels but time-locked to recovery of consciousness (ROC). The red vertical lines indicate the average ± 1 standard deviation time of propofol offset. The black vertical line indicates time zero (ROC). Significant increases or decreases from Awake are opaque colors and are highlighted. PFC: ventrolateral prefrontal cortex; 8A: caudal lateral PFC; PPC: posterior parietal cortex area 7A/7B; STG: superior temporal gyrus.

The online version of this article includes the following figure supplement(s) for figure 7:

**Figure supplement 1.** Effect sizes for change in thalamo-cortical phase synchronization during Awake vs. Unconscious states (left panels).

frequency ranges, but as with cortico-cortical phase synchronization, these effects were smaller (*Figure 7—figure supplement 1*). Uniquely to phase synchronization between thalamus and PFC, we observed a sustained increase in alpha (7–15 Hz) frequency range phase synchrony while Unconscious (p<0.01, cluster-based non-parametric randomization test, *Figure 7A*).

## Thalamic stimulation arouses unconscious monkeys and causes a partial reversal of neurophysiological signs of the Unconscious state

The thalamus is a major route by which ascending excitatory projections from the brainstem reach the cortex. We tested whether we could induce arousal and restore awake-like neurophysiological markers by electrically stimulating the thalamus. During the Unconscious state, we applied 180 Hz, bipolar stimulation targeting the central thalamus, including the mediodorsal nucleus and

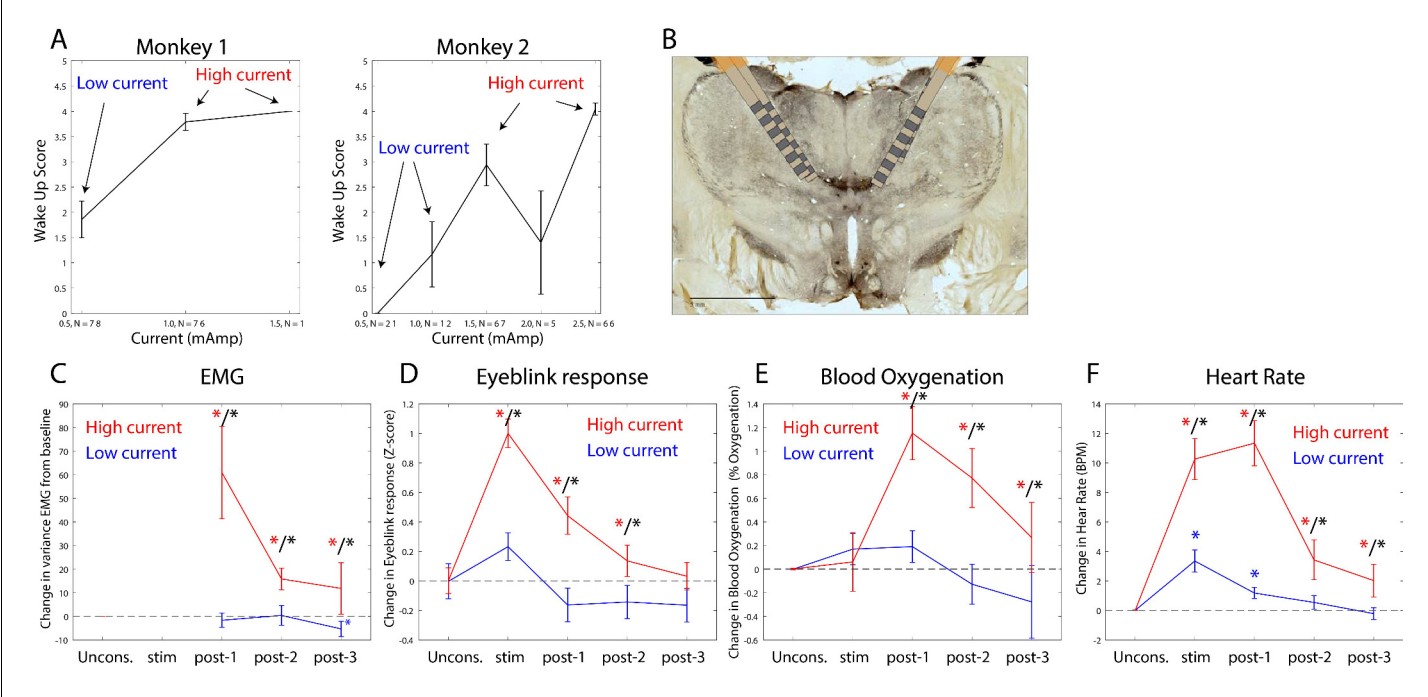

**Figure 8.** Thalamic electrical stimulation in central thalamus arouses monkeys. (**A**) Behavioral wake-up score as a function of thalamic current for monkey 1 (left subplot) and monkey 2 (right subplot). Mean ±+ 1 SEM. A high-current and low-current condition was individually titrated per monkey for producing scores on average above or below a wake-up score of 2. (**B**) A histological image from monkey 1 showing the thalamic leads in the central thalamus. (**C**) Change in EMG from the pre-stimulation Unconscious baseline for high-current (red) vs. low-current (blue) conditions. Change in the physiological signal was tested for difference from Unconscious during the stimulation period (0–28.5 s with respect to electrical stimulation onset), post-1 (0–30 s with respect to electrical stimulation offset), post-2 (30–60 s with respect to electrical stimulation offset), and post-3 (60–90 s with respect to electrical stimulation offset). Significant differences from zero are indicated with red (for high current) or blue (for low current) stars. Significant differences between high vs. low current are indicated with black stars. Mean ± 1 SEM. (**D**) Same as C, but for eyeblink response to air puffs. (**E**) Same as C, but for blood oxygenation. (**F**) Same as C but for the heart rate response.

The online version of this article includes the following figure supplement(s) for figure 8:

**Figure supplement 1.** Effects of thalamic electrical stimulation on wake-up sub-scores.

**Figure supplement 2.** Effects of thalamic electrical stimulation on arousal relative to the Awake state.

intralaminar nuclei (**Figure 8B**, see 'Methods', a minority of sites were in neighboring thalamic nucleus VPL). We applied either high or low current, titrating it for each individual animal (**Figure 8A**) based on the change in arousal that it evoked. We measured arousal with a 'wakeup score' (see 'Methods'). It assessed whether the eyes opened and whether there was an increase in limb movements and puff-evoked eyeblinks (see 'Methods'). A wakeup score of zero indicated none of these happened. A value of 6 indicated that they all occurred. All three sub-scores contributed to the wakeup score. There was a larger contribution from eye-opening and puff-evoked eyeblinks (**Figure 8—figure supplement 1**). During the Unconscious state without electrical stimulation, we did not observe these events (e.g., **Figure 1D** shows that on average there were no air puff-evoked eyeblinks while Unconscious). Analysis of the electromyography (EMG), blood oxygenation, eyeblinks to airpuffs, and heart rate during thalamic electrical stimulation corroborated this wakeup score: Electrical stimulation of the thalamus increased muscle tone (non-parametric randomization test, p<0.01, **Figure 8C**), eyeblinks to airpuffs (non-parametric randomization test, p<0.01, **Figure 8D**), blood oxygen saturation (non-parametric randomization test, p<0.01, **Figure 8E**), and heart rate (non-parametric randomization test, p<0.01, **Figure 8F**). These changes were all greater for high relative to low-current (**Figure 8C–F**, non-parametric randomization test for significant differences between high- vs. low-current stimulation, p<0.01 are indicated with black stars) and outlasted the electrical stimulation period itself, achieving significance during the time window from 0 to 30 s post

stimulation offset (post-stim one in *Figure 8C–F*) and sometimes the 30–60 s (post-stim 2) or 60–90 s (post-stim 3) post stimulation offset periods.

We quantified behavioral and neurophysiological effects relative to the Awake state using a Return To Wakefulness score (RTW – see 'Methods' and *Table 1*). An RTW score of 0% denotes no change and RTW of −100% denotes full return to the Awake state. High-current thalamic stimulation induced an average heart rate increase of 6–8 beats/min above Awake levels and increased eye-blinks to airpuffs and blood oxygenation (RTW of −36% and −69%, respectively) and did not change EMG responses (*Figure 8—figure supplement 2*, *Table 1*).

Stimulation produced an awake-like cortical state by increasing spiking rates and decreasing slow-frequency power and synchronization. An example raster plot of well-isolated single neurons from a single stimulation trial is shown in *Figure 9A*. To see how we removed spurious threshold crossings due to the electrical stimulation itself, see Methods. High-current stimulation increased the spike rate (*Figure 9B* and *Figure 9C*, upper subpanel) from ~1 to ~2.5 spikes/s. In all areas, there was a significant increase in spiking during the stimulation interval compared to the pre-stimulation Unconscious state (*Figure 9C*, orange bars in upper subpanel, red stars indicate significant

**Table 1.** Return to Wakefulness (RTW) scores for effects of high-current thalamic stimulation on physiological measures of arousal, firing rates, and cortical power.

| Measure | Awake | Pre-stim unconscious | Stim | Post-stim1 | Post-stim2 | Post-stim3 |
|---|---|---|---|---|---|---|
| EMG variance | −100% | 0% | −2% (ns) | 0% (ns) | 0% (ns) | 0% (ns) |
| Eyeblinks | −100% | 0% | −36% (*) | −32% (*) | −7% (ns) | −2% (ns) |
| Blood oxygenation | −100% | 0% | −1% (ns) | −69% (*) | −48% (*) | −19% (ns) |
| Heart rate | −100% | 0% | −328% (*) | −362% (*) | −109% (ns) | −64% (ns) |
| Firing rate (PFC) | −100% | 0% | −6% (ns) | −16% (*) | −9% (ns) | −4% (ns) |
| Firing rate (8A) | −100% | 0% | −10% (ns) | −13% (ns) | −8% (ns) | −4% (ns) |
| Firing rate (PPC) | −100% | 0% | −33% (*) | −18% (*) | −11% (ns) | −6% (ns) |
| Firing rate (STG) | −100% | 0% | −26% (*) | −15% (ns) | −6% (ns) | −3% (ns) |
| Slow frequency power (PFC) | −100% | 0% | −45% (*) | −19% (*) | −11% (ns) | −6% (ns) |
| Slow frequency power(8A) | −100% | 0% | −65% (*) | −31% (*) | −18% (*) | −10% (*) |
| Slow frequency power (PPC) | −100% | 0% | −44% (*) | −7% (*) | −1% (ns) | −1% (ns) |
| Slow frequency power (STG) | −100% | 0% | −89% (*) | −8% (ns) | 2% (ns) | 1% (ns) |
| Beta power (PFC) | −100% | 0% | 400% (*) | 1071% (*) | 799% (*) | 449% (*) |
| Beta power (8A) | −100% | 0% | −265% (*) | −228% (*) | −153% (*) | −88% (*) |
| Beta power (PPC) | −100% | 0% | −27% (*) | −29% (*) | −23% (*) | −17% (*) |
| Beta power (STG) | −100% | 0% | −24% (*) | −44% (*) | −31% (*) | −16% (*) |
| Gamma power (PFC) | −100% | 0% | −47% (*) | −48% (*) | −26% (*) | −12% (*) |
| Gamma power (8A) | −100% | 0% | −59% (*) | −40% (*) | −22% (*) | −11% (*) |
| Gamma power (PPC) | −100% | 0% | −71% (*) | −45% (*) | −27% (*) | −15% (*) |
| Gamma power (STG) | −100% | 0% | −77% (*) | −43% (*) | −19% (*) | −6% (ns) |

Asterisks denote significant (p<0.01) changes from the Unconscious state.

PFC: prefrontal cortex; PPC: posterior parietal cortex 7A/7B; 8A: caudal lateral PFC; STG: superior temporal gyrus.

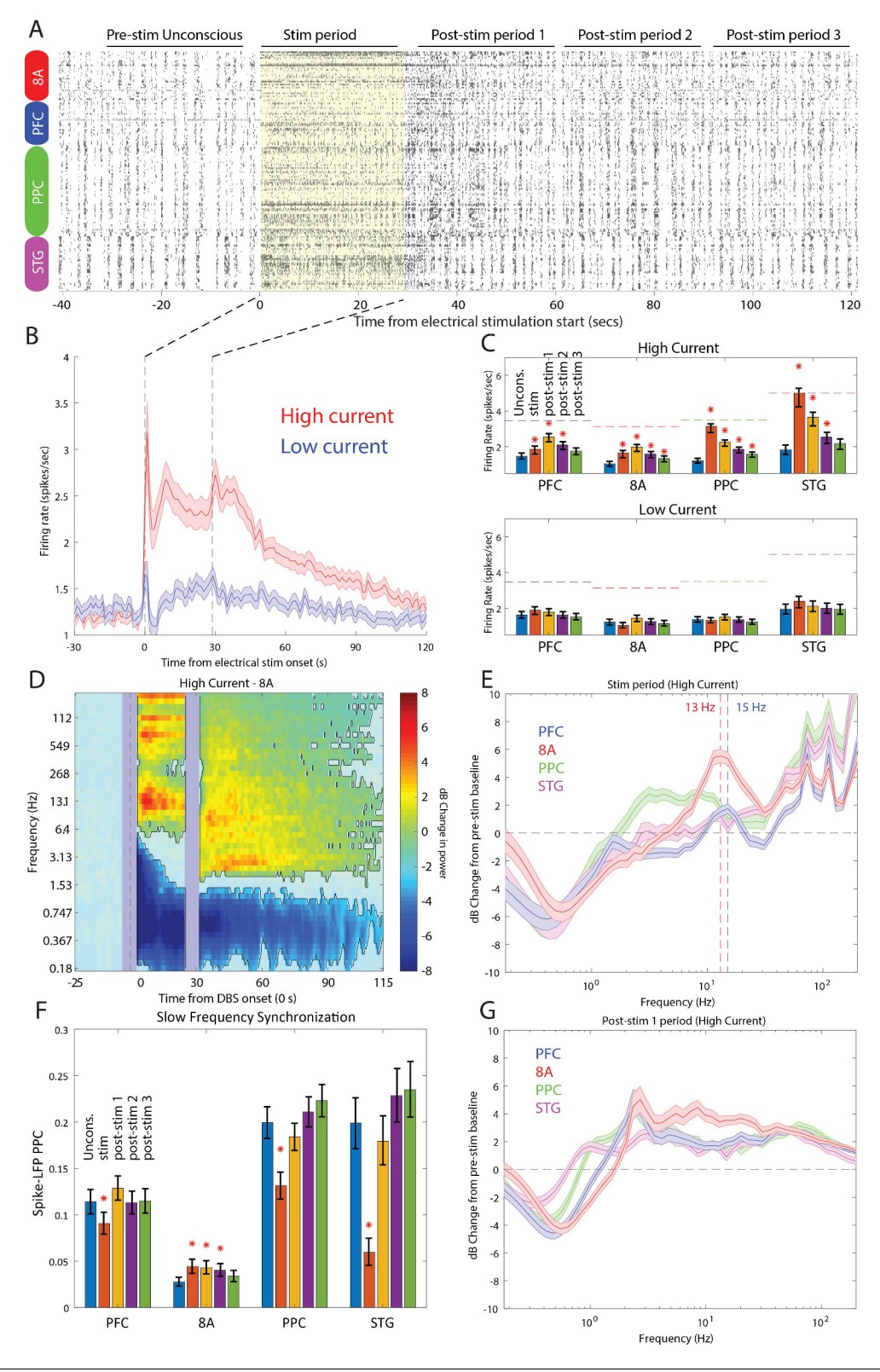

**Figure 9.** Effects of thalamic electrical stimulation on cortical state. (**A**) An example trial. Well-isolated single units are shown before, during, and after electrical stimulation for a trial that produced a maximal wake-up score (eyes opened, muscle movement, response to air puffs). (**B**) The average effect on all well-isolated single units across all areas. Mean firing rates with respect to electrical stimulation onset (at time zero) and offset (28.5 s) for high (red) vs. low (blue) current. (**C**) (Upper panel) Mean firing rates for all single units in each area as a function of time in the trial during high-current

*Figure 9 continued on next page*

*Figure 9 continued*

stimulation (blue bars: pre-stim Unconscious baseline; orange bars: electrical stimulation; yellow bars: 0–30 s after electrical stimulation offset; purple bars: 30–60 s after electrical stimulation offset; green bars: 60–90 s after electrical stimulation offset). (Lower panel) Same as upper subplot, but for low-current stimulation. Mean ± the bootstrap 99 percent confidence interval of the mean across neurons. (D) Mean dB change in power as a function of time since electrical stimulation for all high-current trials in area 8A. (E) Same as D, but highlighting spectral modulation during the period of electrical stimulation (0–28.5 s). Mean ± 1 SEM. The vertical dotted lines indicate the peak frequencies of beta power modulation in PFC and 8A. (F) Mean pairwise phase consistency (PPC) between spikes and fields within each cortical area. Significant changes relative to the pre-stimulation Unconscious state are indicated with asterisks (p<0.01, non-parametric randomization test). Mean and the bootstrap 99 percent confidence interval of the mean across neurons. (G) Same as D, but highlighting spectral modulation during the post-1 period of electrical stimulation (0–30 s post electrical stimulation offset). Mean ±/- 1 SEM. PFC: prefrontal cortex; PPC: posterior parietal cortex 7A/7B; 8A: caudal lateral PFC; STG: superior temporal gyrus.

The online version of this article includes the following figure supplement(s) for figure 9:

**Figure supplement 1.** Effects of high-current thalamic electrical stimulation on cortical firing rates relative to the Awake state.
**Figure supplement 2.** Effects of thalamic electrical stimulation on cortical power.
**Figure supplement 3.** Effects of high-current thalamic electrical stimulation on cortico-cortical synchronization relative to the Awake state.
**Figure supplement 4.** Effects of high-current thalamic electrical stimulation on thalamo-cortical synchronization relative to the Awake state.
**Figure supplement 5.** Effects of high-current thalamic electrical stimulation on SF power relative to the Awake state.
**Figure supplement 6.** Effects of high-current thalamic electrical stimulation on SF spike-LFP synchronization relative to the Awake state.
**Figure supplement 7.** Effects of high-current thalamic electrical stimulation on alpha-beta power relative to the Awake state.
**Figure supplement 8.** Effects of high-current thalamic electrical stimulation on gamma power relative to the Awake state.

differences, p<0.01, non-parametric randomization test). In area STG, this increase brought the average spike rate to the same level as that seen during natural ROC (horizontal dotted lines in *Figure 9C*, upper subpanel). In all areas, the increased spike rate persisted for at least 30 s (and as long as 90 s) post-stimulation (*Figure 9C*, red stars indicate significant differences, p<0.01, non-parametric randomization test). With low-current stimulation, mean spikes rates were unchanged from pre-stimulation baseline (*Figure 9C*, lower subpanel, all p>0.01). Measured relative to the Awake state, high-current thalamic stimulation increased firing rates, with RTW between −17% and −33% (*Figure 9—figure supplement 1* and *Table 1*).

LFP power/synchronization was modulated by electrical stimulation (shown for area 8A in *Figure 9D* and for the other areas in *Figure 9—figure supplement 2*). High-current, but not low-current, stimulation significantly reduced slow-frequency power for up to 85 s post electrical stimulation onset relative to pre-stimulation baseline (*Figure 9D* and *Figure 9—figure supplement 2*, p<0.01, cluster-based non-parametric randomization test). In addition, high-current stimulation decreased within-area spike-LFP (in three of four cortical areas, *Figure 9F*) as well as cortico-cortical and thalamo-cortical LFP-LFP slow-frequency synchronization (*Figure 9—figure supplements 3* and *4*). High-current stimulation also significantly increased higher-frequency power during and after stimulation (*Figure 9D* and *Figure 9—figure supplement 2*, p<0.01, cluster-based non-parametric randomization test). During stimulation, the enhancement had a well-defined peak in the alpha-beta band in area 8A and PFC (vertical dotted lines in *Figure 9E* indicate peaks at 13 Hz in 8A and 15 Hz in PFC). They were absent in the 0–30 s post-stimulation offset interval (*Figure 9G*). The reduction in slow frequency power persisted for 85 s post-stimulation. All of the effects were weak or absent for low-current stimulation (*Figure 9—figure supplement 2*). Measured relative to the Awake state, high-current thalamic stimulation decreased slow-frequency power (RTW: −44 to −89%, *Figure 9—figure supplement 5*, *Table 1*), within-area spike-LFP synchronization (RTW: −24%, −35%, −113%, area 8A showed opposite results, a slight increase in spike-LFP synchronization, *Figure 9—figure supplement 6*, *Table 2*), thalamo-cortical (RTW: −71 to −77%, *Figure 9—figure supplement 4*, *Table 2*) and cortico-cortical synchronization (RTW: −29 to −63%%, *Figure 9—figure supplement 3*, *Table 2*). Beta and gamma power also increased relative to the Awake state. Beta power enhanced beyond that seen in the Awake state in frontal areas (*Figure 9—figure supplement 7*, *Table 1*) and gamma power reached RTW values between −47% and −77% (*Figure 9—figure supplement 8*, *Table 1*).

## Discussion

Our study provides new details on the neurophysiological effects of propofol-induced unconsciousness. Just prior to the Unconscious state, there was an increase in alpha-beta power in frontal cortex.

**Table 2.** Return to Wakefulness (RTW) scores for effects of high-current thalamic stimulation on LFP-LFP and spike-LFP slow-frequency phase synchronization.

| Measure | Awake | Pre-stim unconscious | Stim | Post-stim1 | Post-stim2 | Post-stim3 |
|---|---|---|---|---|---|---|
| Spike-LFP (PFC) | −100% | 0% | −24% (*) | 15% (ns) | −1% (ns) | 1% (ns) |
| Spike-LFP (8A) | −100% | 0% | 103% (*) | 96% (*) | 79% (*) | 38% (ns) |
| Spike-LFP (PPC) | −100% | 0% | −35% (*) | −8% (ns) | 6% (ns) | 12% (*) |
| Spike-LFP (STG) | −100% | 0% | −113% (*) | −16% (ns) | 24% (ns) | 29% (ns) |
| LFP-LFP (PFC-thalamus) | −100% | 0% | −77% (*) | −58% (*) | −7% (ns) | 8% (ns) |
| LFP-LFP (8A-thalamus) | −100% | 0% | −77% (*) | −71% (*) | −16% (ns) | −4% (ns) |
| LFP-LFP (PPC-thalamus) | −100% | 0% | −10% (ns) | −19% (ns) | 30% (ns) | 15% (ns) |
| LFP-LFP (STG-thalamus) | −100% | 0% | −71% (*) | −68% (*) | −27% (*) | −12% (ns) |
| LFP-LFP (PFC-8A) | −100% | 0% | −40% (*) | −25% (*) | −6% (ns) | 3% (ns) |
| LFP-LFP (PFC-PPC) | −100% | 0% | −60% (*) | 3% (ns) | 17% (ns) | 1% (ns) |
| LFP-LFP (PFC-STG) | −100% | 0% | −45% (*) | −11% (ns) | 6% (ns) | 36% (*) |
| LFP-LFP (8A-PPC) | −100% | 0% | −29% (*) | −13% (ns) | −7% (ns) | 8% (ns) |
| LFP-LFP (8A-STG) | −100% | 0% | −63% (*) | −41% (*) | −8% (ns) | 11% (ns) |
| LFP-LFP (PPC-STG) | −100% | 0% | 29% (*) | 5% (ns) | 9% (ns) | 4% (ns) |

Asterisks denote significant (p<0.01) changes from the Unconscious state.

PFC: prefrontal cortex; PPC: posterior parietal cortex 7A/7B; 8A: caudal lateral PFC; STG: superior temporal gyrus; LFP: local field potentials.

By contrast, alpha-beta power in posterior cortex transiently increased and then decreased relative to the awake state. During the Unconscious state, there was a widespread decrease in high-frequency (greater than 50 Hz) power and an increase in slow-frequency (~1 Hz and lower) power. Spike rates decreased and spiking became coupled to the slow-frequency oscillations. There was an increase in slow-frequency cortico-cortical and thalamo-cortical phase synchronization. In frontal cortex alone, there was an increase in alpha-beta (~8–30 Hz) power and cortico-cortical and cortico-thalamic synchronization. These effects were distributed differently across cortical layers. In superficial layers (2/3), there was a stronger suppression of gamma and spiking during the Unconscious state. Deep layers (5/6) showed a stronger increase in slow-frequency power.

Our results agree with results in humans that have related slow-frequency oscillations with propofol-induced unconsciousness. We also observed a profound decrease in cortical spiking and an increase of its phase-locking with slow-frequency oscillations. We also saw anteriorization of the alpha-beta oscillations. Any or all of this may cause unconsciousness by disrupting cortical communication, especially in the theta, beta, and gamma ranges associated with cognition and consciousness (*Bastos et al., 2015*; *Brown et al., 2010*; *Buschman and Miller, 2007*; *Cimenser et al., 2011*; *Fiebelkorn and Kastner, 2019*; *Lakatos et al., 2008*; *Lewis et al., 2012*; *Miller et al., 2018*; *Vijayan et al., 2013*). The evidence that propofol fragmented cortex is less well supported. We also found different effects across layers that support other theories. *Dehaene and Changeux, 2011* proposed that the conscious state relies on 'broadcasting' of cortical activity throughout cortex via long-range superficial connections. *Aru et al., 2019* suggested that consciousness relies on cortico-thalamic broadcasting which depends on integrity of sub-cortical projection neurons in deep layers.

The thalamus is known to contribute to cortical dynamics in the Awake state (*Fiebelkorn et al., 2019*; *Fiebelkorn and Kastner, 2019*). The ILN of the central thalamic nucleus has diffuse cortical connections posited to mediate the global binding needed for consciousness (*Llinás et al., 1998*). Thalamic stimulation has improved behavioral performance of a minimally conscious patient (*Schiff et al., 2007*). Correspondingly, we found that stimulation of the central thalamus caused monkeys to regain arousal, similar to a recent report (*Redinbaugh et al., 2020*). The stimulation

increased cortical spike rates, diminished slow-frequency power and synchronization, and reinstated higher-frequency power.

Our results are consistent with rodent studies of recovery from anesthesia induced by stimulation. This includes intravenous administration of stimulants (*Chemali et al., 2012*; *Kenny et al., 2015*; *Kenny et al., 2015*; *Solt et al., 2011*; *Taylor et al., 2013*), site-specific electrical stimulation (*Muindi et al., 2016*; *Pillay et al., 2014*; *Solt et al., 2014*), and site-specific optogenetic stimulation (*Taylor et al., 2016*).

During loss of consciousness, alpha-beta (~8–30 Hz) power increased in frontal areas and decreased in posterior areas. Thalamic stimulation also resulted in an increase in alpha-beta power that was particularly pronounced in frontal areas. Previous studies have documented that synchronization between cortex and thalamus is prominent in the alpha-beta frequency range (*Bastos et al., 2014*; *Fiebelkorn et al., 2019*; *Saalmann et al., 2012*). We propose that alpha-beta is this circuit's natural resonance frequency. We enhanced it by driving the thalamo-cortical loop via thalamic stimulation. Because frontal alpha-beta power was already enhanced during unconsciousness relative to the pre-drug Awake state, we do not suppose that the enhanced beta was contributing to the increased arousal we observed.

Here, we focused on propofol's actions on cortex and thalamus. The mechanisms we suggest are likely relevant for other anesthetics that target GABA receptors (e.g., sevoflurane, isoflurane, desflurane, barbiturates, and etomidate). Future studies will be needed to investigate the extent to which propofol's GABA-ergic targets in the brainstem contribute to unconsciousness (*Brown et al., 2018*; *Brown et al., 2011*; *Brown et al., 2010*). Anesthetics that target NMDA antagonists such as ketamine and nitrous oxide have different dynamics and mechanisms (*Akeju et al., 2016*; *Pavone et al., 2016*; *Purdon et al., 2015*). We have interpreted the neurophysiological changes as the cause of unconsciousness. It is important to note that there were other factors, including oxygen saturation, blood pressure, and ventilation, which differed between the Awake and Unconscious states in our study that may also be contributory. We were not able to separate these factors vs. the observed changes in neurophysiology to producing states of unconsciousness.

A recent study (*Redinbaugh et al., 2020*) also found that propofol reduced alpha and gamma cortical synchrony and that thalamic stimulation aroused the NHPs and increased alpha-gamma synchrony. But there were some differences between their study and ours. *Redinbaugh et al., 2020* found less of a contribution of slow frequencies to the unconscious state and less slow-frequency modulation due to thalamic stimulation. They found decreases in spike rates only in deep cortical layers. We found a stronger contribution of slow frequencies and decreases in spiking in all layers (with a stronger decrease in superficial layers). *Redinbaugh et al., 2020* administered ketamine before propofol whereas we administered propofol alone. They compared Awake to Unconscious in separate sessions. We studied the transition in and out of consciousness within a session. There were also differences in thalamic stimulation. They used a single stimulating linear array centered in the ILN and found that 50 Hz was most effective at inducing arousal from the unconscious state. By contrast, we used larger stimulating electrode arrays in pairs targeting both ILN and MD, higher currents, and 180 Hz. We did so to approximate stimulation parameters that were effective in increasing arousal in a minimally conscious human (*Schiff et al., 2007*).

In sum, a mechanism through which propofol likely renders unconsciousness is by disrupting intra-cortical and thalamo-cortical communication through decreased spiking and enhanced slow-frequency power/synchrony. This leads to low-spiking Down-states and loss of the higher-frequency coherence thought to integrate cortical information (*Baars et al., 2013*; *Cimenser et al., 2011*; *Crick and Koch, 1990*; *Dehaene and Changeux, 2011*; *Dehaene, 2001*; *Demertzi et al., 2019*; *Engel et al., 1999*; *Lee et al., 2013*; *Purdon et al., 2013*; *Tononi et al., 2016*).

## Materials and methods

### Experimental subjects and vascular access port

Four rhesus macaques (*Macaca mulatta*) aged 14 years (monkey 1, male, ~13.0 kg), 8 years (monkey 2, female, ~6.6 kg), 7 years (monkey 3, female, ~5.0 kg), and 18 years (monkey 4, female, ~11,9 kg) participated in these experiments. All animals were pair-housed on 12 hr day/night cycles and maintained in a temperature-controlled environment (80 °F). Monkeys 1 and 2 were surgically implanted

with a subcutaneous vascular access port (Model CP-6, Norfolk Access Technologies, Skokie, IL) at the cervicothoracic junction of the neck with the catheter tip reaching the termination of the superior vena cava via the external jugular vein. Monkeys 3 and 4 were acutely implanted with a catheter to a vein in the ear after applying lidocaine to the overlying skin. The catheter was removed and reapplied prior to each session.

## Neural recordings in cortex

Neurophysiology with chronic Utah arrays: For recordings in cortex, monkeys 1 and 2 were chronically implanted with four $8 \times 8$ iridium-oxide contact microelectrode arrays ('Utah arrays', MultiPort: 1.0 mm shank length, 400 µm spacing, Blackrock Microsystems, Salt Lake City, UT), for a total of 256 electrodes. Arrays were implanted in the prefrontal (area 46 ventral and 8A), posterior parietal (area 7A/7B), and temporal-auditory (caudal parabelt area STG [superior temporal gyrus]) cortices. Specific anatomical targeting utilized structural MRIs of each animal and a macaque reference atlas, as well as visualization of key landmarks on surgical implantation (*McLaren et al., 2009*). For Utah array recordings, area 8A and PFC were ground and referenced to a common subdural site. Area STG and PPC also shared a common ground/reference channel which was also subdural. LFPs were recorded at 30 kHz and filtered online via a lowpass 250 Hz software filter and downsampled to 1 kHz. Spiking activity was recorded by sampling the raw analog signal at 30 kHz, bandpass filtered from 250 Hz to 5 kHz, and manually thresholding. Blackrock Cereplex E headstages were utilized for digital recording via 2–3 synchronized Blackrock Cerebus Digital Acquisition systems. Single units were sorted manually offline using principal component analysis with commercially available software (Offline Sorter v4, Plexon Inc, Dallas, TX). All other pre-processing and analyses were performed with Matlab (The Mathworks, Inc, Natick, MA).

To ensure signals recorded on the multiple data acquisition systems remained synchronized with zero offset, a synchronization test signal with locally unique temporal structure was recorded simultaneously on one auxiliary analog channel of each system. The relative timing of this test signal between each system's recorded datafile was measured offline at regular intervals throughout the entire recording session. Any measured timing offsets between datafiles were corrected by appropriately shifting spike and event code timestamps in time, and by linearly interpolating analog signals to a common time base.

Neurophysiology with acute laminar probes: For recordings in monkeys 3 and 4, the monkeys were first implanted with a custom-machined Carbon PEEK chamber system with three recording wells placed over visual/temporal, parietal, and frontal cortex (*Mulliken et al., 2015*). We acutely introduced 16 or 32 contact 'multi-laminar' probes (U/V probes, Plexon, Dallas, TX) into the same cortical areas we recorded with chronic Utah arrays: areas STG, 7A/7B, 8A, and VLPFC. Between 1 and 2 probes were used per recording chamber and a total of 4–6 probes were used per session. The total channel count ranged between 96 and 128 electrodes per session. Electrode contacts on these probes were spaced 100 µm apart for the 32 channel probes or 200 µm apart for the 16 channel probes. This gave a total linear sampling of 3.0–3.1 mm on each probe. The recording reference was the reinforcement tube, which made metallic contact with the entire length of the probe (total probe length from connector to tip was 70 mm). With MRI guidance, we introduced the probes to be perpendicular to cortex and to span all cortical layers, as previously described (*Bastos et al., 2020*). As a marker for layer 4, we used the relative power profiles calculated in the pre-propofol Awake state. The cross-over between the relative power of the gamma and alpha-beta bands was used to estimate the location of layer 4. This cross-over point between gamma and alpha-beta relative power profiles was previously shown to correspond to the location of the first current source density sink (within 100–200 µm), another marker for layer 4 (*Bastos et al., 2018*).

## Neural recordings and electrode targeting in thalamus

After 10–20 sessions of cortex only recordings, monkeys were chronically implanted with four 6–8 channel recording/stimulating electrodes (0.5 mm contacts with 0.5 mm intercontact spacing, NuMed Inc, Hopkington, NY) bilaterally targeting the intralaminar nuclei of the central thalamus. Thalamic recordings were referenced to the monkey's titanium headpost. Thalamic sites were re-referenced to a bipolar montage prior to phase synchronization analysis. LFPs were recorded via a separate analog front-end amplifier and an additional identical digital acquisition system, synchronized

to the digital acquisition systems utilized for cortical recordings. LFPs were similarly recorded at 30 kHz and filtered online via a lowpass 250 Hz software filter and downsampled to 1 kHz.

A specialized anatomical localization and insertion protocol utilizing serial intraoperative MRIs was developed in order to allow for precise subcortical targeting of the electrodes along the long axis of the central lateral nucleus and extending ventrally to the centromedian and parafasicular nuclei. Custom-made carbon PEEK recording chambers were affixed to the skull with acrylic and ceramic screws stereotaxically determined to target the central thalamus. Recording chamber grids with 1 mm grid holes were inserted into the chambers, filled with sterile saline, and the monkeys were imaged by 3T MRI. After confirmation of the appropriate grid holes targeting the thalamic structures of interest, monkeys were generally anesthetized and brought to the operating facility where a small-bore craniotomy (<2 mm) was performed at the appropriate grid holes. The grid was replaced in the chamber and the monkeys were transferred to the imaging facility under general anesthesia. They were then administered a gadofosveset trisodium contrast agent to highlight vasculature obstructing the trajectory to thalamus (e.g., thalamostriate vein). In the MRI suite, a stylette cannulae was inserted into the relevant grid holes and lowered several millimeters into cortex and one set of 0.5 mm resolution images was obtained. Upon confirmation of correct trajectory on MRI, the stylette-cannulae were lowered to their final position, with the tip approximating the thalamus. The stylettes were then removed, and electrodes of marked length lowered to the depth of the cannulae. Following another MRI-based measurement (scan 2) with the electrodes still in the cannulae, the electrodes were lowered to their final positions within the thalamus and reimaged (scan 3). Upon final assessment of correct localization, the probes were fixed in place and the chamber sealed with acrylic. Histological staining with acetylcholinesterase was used to confirm exact electrode contact locations within and outside the central thalamus of both monkeys. Thalamic leads used for recording/stimulating were centered on the ILN (consisting of nuclei CL, Cm, and Pf) and the MD nucleus of the thalamus. A minority of the leads also encompassed neighboring thalamic area VPL.

## Experimental procedures

On a given experimental session, monkeys were head-fixed via a titanium headpost and placed in noise isolation chambers with masking white noise (50 dB). We ran two sets of experimental sessions. The first set of sessions consisted of neurophysiological recordings from cortex only. We refer to these as 'propofol cortex sessions.' A total of 21 sessions (N = 11 from monkey 1, N = 10 from monkey 2) were used. These sessions proceeded as follows: first, a period of 15–90 min of awake baseline electrophysiological recordings were recorded. Next, propofol was intravenously infused via a computer-controlled syringe pump (PHD ULTRA 4400, Harvard Apparatus, Holliston, MA). The infusion protocol was stepped such that unconsciousness was induced via a higher rate infusion (285 µg/kg/min for monkey 1; 580 µg/kg/min for monkey 2) for 20 min before dropping to a maintenance dose (142.5 µg/kg/min for monkey 1; 320 µg/kg/min for monkey 2) for an additional 40 min.

The second set of experimental sessions occurred after we implanted the thalamic recording/stimulation electrodes (deep brain stimulating electrodes [DBS]). We refer to these as the 'thalamus wakeup sessions.' After the initial 20 min of propofol infusion, DBS stimulation trials began. A total of 22 sessions (N = 11 from monkey 1, N = 11 from monkey 2) were analyzed with thalamic stimulation.

Heart rate and oxygen saturation were monitored continuously and recorded throughout all phases of experiments using clinical-grade pulse oximetry (Model 7500, Nonin Medical, Inc, Plymouth, MN). SpO$_2$ values were maintained at values above 93% for each of the recording sessions.

Infrared monitoring tracked facial movements and pupil size (Eyelink 1000 Plus, SR-Research, Ontario, CA) throughout the course of the experiments. Loss of consciousness (LOC) was deemed by the timestamp of the moment of eyes-closing that persisted for the remainder of the infusion. ROC was classified as the timestamp of the first to occur between eyes reopening or regaining of motor activity following drug infusion cessation. Animals regained consciousness after the maintenance infusion was terminated and were monitored for an additional period before being returned to their home cage. To ensure propofol clearance from tissues and physiological recovery, experiments were never repeated on subsequent days. All procedures followed the guidelines of the MIT Animal Care and Use Committee (protocol number 0619-035-22) and the US National Institutes of Health.

## Thalamic stimulation procedure

For electrical stimulation of thalamic electrodes, we adapted electrical stimulation parameters previously shown to cause behavioral improvements in coma patients and awake, behaving monkeys (*Baker et al., 2016*; *Schiff et al., 2007*). We unilaterally delivered 180 Hz bipolar, biphasic, square wave pulses (0.5–2.5 milliAmps) between 6 and 8 contacts on monkey 1 and 6 contacts on monkey 2. Stimulation montages were used that included two thalamic probes on the same side. Five minutes into the maintenance anesthesia dose (20 min from infusion start), 28.5 s 'trials' of electrical stimulation were delivered as the propofol infusion continued. DBS trials were separated by 2 min intervals, except the 4th and 5th stimulation runs, which were separated by a 5 min inter-trial interval. These DBS washout periods sufficiently allowed for reestablishment of the behaviorally judged unconscious state (e.g., loss of puff responses). We delivered between 0.5 and 2.5 mAmp of current. In early pilot experiments, different currents, frequencies, waveform shapes, and electrode combinations were screened for eliciting arousal. We chose a set of parameters that was effective at eliciting arousal, but minimal in current and number of stimulated electrodes, for the final experiments reported here.

## Wakeup score

One of the authors (MM) who was present during all DBS sessions performed a behavioral score for each DBS trial for the degree to which the electrical stimulation induced changes in arousal. This numerical 'wakeup score' is loosely inspired by the Glasgow coma scale. Like the Glasgow coma scale, it separately scores each of the behavioral components, then sums them into a single overall score for each DBS trial. The components are: (1) spontaneous eye opening (0–2): 0 = eyes closed, 1 = one or both eyes slightly open, 2 = one or both eyes fully open; (2) responses to external stimuli (airpuffs directed at eye/face; 0–2): 0 = no response, 1 = occasional blinking not necessarily in response to puffs, 2 = clear response to airpuffs; (3) face/body movements (0–2): 0 = none, 1 = some mouth/jaw movement, 2 = arm/full-body movement. The final waking score for each trial is simply the sum of these three components. Note that the actual Glasgow scale combines components 1 and two into a single 'eye opening' score, but empirically in this data, spontaneous eye opening and blink responses to airpuffs seem to occur somewhat independently rather than on a single continuum.

## Histology

At the end of the recording sessions, the animals were euthanized for histological confirmation of thalamic sites, as previously described (*Wu and Kaas, 1999*). Briefly, monkeys were given a lethal dose of sodium pentobarbital. When they became areflexive, they were perfused transcardially with PBS, followed first by a cold solution of 4% paraformaldehyde and next by a mixed solution of 4% paraformaldehyde and 10% sucrose. Blocks of brain and spinal cord were removed and stored overnight in 30% sucrose at 5℃ before cutting. Sections were processed for acetylcholinesterase. Anatomical localization of electrodes was determined by histological examination of brain tissue. It was not necessary to create electrolytic lesions prior to histology, because the thalamic electrodes were wide enough (0.5 mm diameter) to be unambiguously identified in the anatomical sections.

## Data preprocessing, general

Single units were sorted manually offline using principal component analysis with commercially available software (Offline Sorter v4, Plexon Inc, Dallas, TX). All other pre-processing and analyses were performed with Matlab (The Mathworks, Inc, Natick, MA).

## Data preprocessing, electrical stimulation LFP

Electrical stimulation generally produced artifacts that were highly correlated across channels in a stereotyped manner. We removed these using zero-phase component analysis (ZCA) whitening (*Eldar and Oppenheim, 2003*). ZCA whitening is the linear whitening transformation that minimizes the mean squared error between the original and whitened signals. Intuitively, it estimates the across-channel correlations induced by DBS stimulation, and removes them from the data. First, we extracted the spiking band from the raw 30 kHz analog signals by band-pass filtering at 300–4500 Hz with a zero-phase fourth-order Butterworth filter. Next, we estimated the cross-channel

covariance matrix Σ from the filtered signals during the DBS stimulation periods (from a subset of 100,000 randomly sampled time points, for computational efficiency). From the estimated covariance, we computed the 'whitening matrix': $W = \Sigma^{-1/2}$. We then normalized each column of $W$ by its diagonal (variance) value, so that the resulting matrix will remove DBS-induced correlations, but not change the amplitude of individual channels. We removed the DBS stimulation artifacts by multiplying the full matrix of filtered signals by the modified whitening matrix. Finally, we computed the mean and standard deviation of each denoised channel, thresholded each at –4.5 SD, and extracted spike timestamps and waveforms around each threshold crossing. Extracted spikes were sorted into units using principal component analysis in commercially available software (Offline Sorter v4, Plexon Inc, Dallas, TX). Only spikes whose average waveform during electrical stimulation matched the waveform outside periods of electrical stimulation (Pearson correlation coefficient greater than or equal to 0.99) were included for analysis.

## Methods, LFP spectral analysis, and statistics

LFPs were analyzed using the Fieldtrip toolbox (http://www.fieldtriptoolbox.org/) (*Oostenveld et al., 2011*). We tested whether specific oscillations in different areas relative to drug onset (or electrical stimulation onset) were modulated in power. For each channel on each array or thalamic probe, we computed a time-frequency decomposition. For propofol-only sessions, we calculated power in sliding windows of duration 10 s with a hanning taper, to deliver 0.1 Hz Spectra resolution. We calculated power across logarithmically spaced frequencies 0.178–200 Hz.

We calculated the change in power between Unconscious and Awake baseline in decibel (dB) units. In other words, we applied the following transformation to the raw power values:

$$10 \log_{10} \left( \frac{power\, Unconscious}{power\, Awake} \right)$$

For cortex power modulation with thalamic electrical stimulation, we calculated power using a sliding window approach time locked to the onset of electrical stimulation. Power was calculated from 25 s pre-stimulation to 150 s post-stimulation with 0.5 s intervals, and with 5 s analysis windows. We calculated change in power relative to pre-stimulation baseline, which was the average power from 25 s to 3 s prior to stimulation.

## Methods, Pairwise Phase Consistency

To quantify phase synchronization between cortico-cortical and thalamo-cortical LFPs, and between spike-field LFP pairs, we used the pairwise phase consistency (PPC) (*Vinck et al., 2010*). The PPC metric is a metric of phase synchronization that is unbiased by the number of observations. Prior to calculating PPC, we first re-referenced data to a local bipolar montage at 1600 μm distance. We then calculated PPC between inter-areal bipolar sites using the multitaper method with 1 Hz spectral smoothing to estimate power in 0.5 Hz intervals from 0 to 200 Hz using 2 s windows with 75% overlap. For thalamo-cortical PPC analysis, we re-referenced cortical recordings as stated above, and we re-referenced thalamus data to a local bipolar montage at 1500 μm distance. For spike-LFP analysis, we average LFPs across each cortical array (STG/PPC/8A/PFC) and computed PPC between these array-averaged LFPs and each well-isolated single neuron in each area.

## Statistics

We determined whether there were differences from baseline in power and coherence by using a non-parametric cluster-based randomization test (*Oostenveld et al., 2011*). For each session, we realized the null hypothesis that power in the baseline and power in the drug period were the same. To this end, we randomly exchanged baseline-transformed time-frequency power estimates between the baseline and drug windows. We extracted the largest cluster (continuous tiles in time-frequency space) to pass a first level significance threshold, by applying a t-test and thresholding all significant bins p<0.01, uncorrected. We performed this randomization 1000 times. The empirically observed clusters were compared to this randomization distribution to assess significance at p=0.01, adjusted for multiple comparisons across sessions.

To calculate the effect of electrical thalamic stimulation on cortical power, we applied a similar transformation, only this time the baseline was the 30 s of data immediately preceding each trial of stimulation:

$$10\log_{10}\left(\frac{power\,stimulation}{power\,pre-stimulation\,Unconscious\,baseline}\right)$$

We then repeated the same test outlined above, only now randomizing bins before/during/after stimulation onset to create the null hypothesis. There were stimulation onset and offset artifacts. We removed the times around onset/offset ± 3 s prior to performing this randomization test. As a result, these artifact times are omitted from the analysis and figure.

### Return to wakefulness metric

We quantified how much thalamic stimulation changed physiological and neural variables using a simple metric, called the return to wakefulness (RTW) metric. We first normalized each physiological or neural variable with respect to the pre-drug Awake state by dB change (for power) or subtraction. On this normalized signal we then computed the percent change from the pre-stimulation to the stimulation, post-stim1, post-stim2, and post-stim3 periods. Therefore, RTW of 0% indicates no change and RTW of −100% denotes return to that signal's level observed during the pre-drug Awake state. Note that a positive RTW value would mean that particular measure moved further away from the levels seen in the Awake state, and that thalamic stimulation moved the measure further in the direction observed in the Unconscious state.

### Methods, state-space modeling of physiological signals

To characterize physiologically the transition from consciousness to unconsciousness, we measured heart rates, muscle tone with EMG, and blood oxygenation. These signals were pre-processed in the following manner. Heart rates and blood oxygenation (SPO$_2$) signals were averaged using non-overlapping windows of 1 s. The EMG signals were z-scored by subtracting its mean and dividing by its session-wide standard deviation. Its variance was computed using one second, non-overlapping windows. From EMG signals measured from two electrodes adjacent to the eyes, we extracted evoked puff responses (eyeblinks) using the following procedure. We computed the average of the EMG signal within a 50–150 millisecond time window, following an air-puff stimulus, for each individual session. We subtracted the mean and divided this time-series by its session-wide standard deviation. Then, we computed a moving average over a 30 s window to obtain a continuous estimate of this response.

To quantify the change in these physiological signals with the administration of propofol, we used a state-space model analysis (*Shumway and Stoffer, 1982*). We assumed that the temporal structure in the log of the EMG variance, the blood oxygen saturation levels, the eyeblinks and the heart rate can be represented as a linear Gaussian state-space model of the form:

$$Observation\,Equation$$

$$y_t^j = z_t + \varepsilon_t^j$$

$$State\,Equation$$

$$z_t = z_{t-1} + v_t,$$

where $y_t^j$ is the physiological signal at time $t$ for session $j = 1, \ldots, J$, $z_t$ is the state at time t, $\varepsilon_t^j$ is independent Gaussian noise with zero mean and variance $\sigma_\varepsilon^2$, and $v_t$ is independent Gaussian noise with zero mean and variance $\sigma_v^2$ for $t = 1, \ldots, T^j$. Here, $J$ is the total number of sessions and $T^j$ is the total number of samples reorded during the $j^{th}$ session. The state-space model was fit to each physiological signal time series using the Expectation-Maximization algorithm (*Shumway and Stoffer, 1982*, *Dempster et al., 1977*).

To compare these physiological measurements across different time points, we computed the probability that a measurement at time $t$ was lower than measurements at all previous time points.

We performed this comparison since these measurements seem to decrease after the first propofol infusion. We computed this probability for all time points using a Monte Carlo algorithm detailed in *Smith et al., 2005*. We considered a result to be statistically significant if the posterior probability was greater than 0.99.

## Testing whether spikes become coupled to the LFP slow-delta oscillation phase

Using a point process generalized linear model (PPGLM), we modeled spike trains of individual neurons as a function of the LFP phase of the slow-delta oscillation (0.3-3Hz) and the neuron's spike history using a logistic link function. (*McCullagh and Nelder, 1989*; *Truccolo et al., 2005*). That is, we represented $\lambda(t|\phi_t, H_t)$, the conditional intensity function (instantaneous spike rate function), as

$$\log\left[\frac{\lambda(t|\phi_t, H_t)}{1 - \lambda(t|\phi_t, H_t)}\right] = \sum_{j=1}^{J}\beta_h^j\eta_{t-j} + \sum_{k=1}^{K}\beta_\phi^k I_{\phi_t}^{\phi_k},$$

where $\sum_{j=1}^{J}\beta_H^j\eta_{t-j}$ defines $H_t$, the spike history going back $J$ time bins, where $\eta_t$ is a 1 if there is a spike in time bin $t$ and is 0 otherwise. We define the effect of the slow-delta oscillation phase as $\sum_{k=1}^{K}\beta_\phi^j I_{\phi_t}^{\phi_k}$, where $I_{\phi_t}^{\phi_k}$ is an indicator function which is 1 if $\phi_t$, the slow-delta phase at time $t$ equals $\phi_k$ and is 0 otherwise. The $\phi_k$'s are $K$ evenly spaced phases of the slow-delta oscillation between $(-\pi, \pi)$. Phase bins were computed using the following procedure. First, we band-passed the LFP in the slow-frequency/delta band (0.3–3 Hz). We then computed the Hilbert transform to extract a measure of the continuous analytic phase. Finally, we binned these phase estimates into 10 linearly spaced bins ranging from $-\pi$ to $\pi$.

History bins were defined as $[1, 2, \ldots, 10, 11-15, 16-20, 21-30, \ldots, 91-100, 101-150, \ldots, 451-500]$ millisecond (ms) bins. The rationale was that immediate history bins (1–10 ms) reflect the neuron's short timescale biophysics, such as its absolute and relative refractory periods and that longer-range history terms reflect network-wide dynamics (*Czanner et al., 2008*). For this reason, immediate history bins increase by 1 ms and longer-range history bins (>100 ms) increase by 50 ms. The number of history terms for each model was chosen using Akaike's Information Criterion (*Akaike, 1974*). The model was fit to the spike train of each neuron using custom software for performing regression with Truncated Regularized Iteratively Reweighted Least Squares (*Komarek and Moore, 2005*). Goodness of fit was assessed using Kolmogorov-Smirnov (K-S) test based on the time-rescaling theorem (*Brown et al., 2002*).

## Testing whether there is an increase in phase-modulation during the unconscious state

We fit GLMs to 10 min of data during the pre-drug awake state and to 10 min of data after LOC. Neurons that did not spike during either periods were not included. To quantify the contribution of phase to predicting the spiking propensity, we computed the SNR with respect to phase ($SNR_\phi$) for each condition (*Czanner et al., 2015*)

$$SNR_\phi = \frac{Dev(n, \beta_H) - Dev(n, \beta) - dim(\beta_H) + dim(\beta)}{Dev(n, \beta) - dim(\beta)},$$

where $Dev(n, \beta)$ denotes the deviance of the PPGLM model with phase and history, $Dev(n, \beta_H)$ is the deviance of the fit of the PPGLM model with the history terms, $dim(\beta)$ is the dimension of the parameters in the phase and history PPGLM model and $dim(\beta_H)$ is the dimension of the parameters in the history only PPGLM model. . We next computed the logarithm to the base 10 of the difference between the SNR for phase of unconsciousness and the SNR for phase of the awake state defined as

$$\Delta SNR_\phi = 10\log_{10}\left(\frac{SNR_\phi^{unconscious}}{SNR_\phi^{awake}}\right).$$

We considered a neuron to exhibit an increased phase modulation during the unconscious state if its

$\Delta\mathrm{SNR}_\phi$ was positive. Using 1320 neurons for PFC, 754 for 8A, 1058 for PPC, and 573 for STG across 10 sessions for NHP 2 and 11 sessions for NHP 1, we counted the number of neurons with an increased phase modulation for each region. We then computed the posterior probability of an increase in phase-modulation using the beta-binomial model and 10,000 Monte Carlo samples (*DeGroot and Schervish, 2012*; *Solt et al., 2011*). We assumed a binomial model as the likelihood function for the proportion of phase-modulated neurons. We used a uniform prior on the interval (0, 1) as the prior density, and a beta posterior density due to conjugacy. We considered a result to be statistically significant based on the posterior probability if this value was greater than 0.99.

## Acknowledgements

We would like to thank Simon Kornblith for help with data analysis. These studies were supported by NIMH R01MH11559, NIGMS P01GM118269, and The JPB Foundation.

## Additional information

### Funding

| Funder | Grant reference number | Author |
| --- | --- | --- |
| National Institute of Mental Health | R01MH11559 | Earl K Miller<br>Emery N Brown |
| National Institute of General Medical Sciences | P01GM118269 | Emery N Brown<br>Earl K Miller |
| The JPB Foundation | | Earl K Miller<br>Emery N Brown |

The funders had no role in study design, data collection and interpretation, or the decision to submit the work for publication.

### Author contributions

André M Bastos, Formal analysis, Investigation, Writing - original draft, Writing - review and editing; Jacob A Donoghue, Formal analysis, Investigation, Writing - original draft; Scott L Brincat, Formal analysis, Methodology; Meredith Mahnke, Mikael Lundqvist, Investigation; Jorge Yanar, Josefina Correa, Ayan S Waite, Formal analysis; Jefferson Roy, Investigation, Methodology, Writing - review and editing; Emery N Brown, Conceptualization, Funding acquisition, Formal analysis, Supervision, Writing - original draft, Project administration, Writing - review and editing; Earl K Miller, Conceptualization, Supervision, Funding acquisition, Writing - original draft, Project administration, Writing - review and editing

### Author ORCIDs

André M Bastos https://orcid.org/0000-0003-1804-4418
Josefina Correa https://orcid.org/0000-0002-5721-5024
Emery N Brown https://orcid.org/0000-0003-2668-7819
Earl K Miller https://orcid.org/0000-0002-0582-6958

### Ethics

Animal experimentation: The animals were handled in accord with National Institutes of Health guidelines and approved by the Massachusetts Institute of Technology Committee on Animal Care (protovcol number: 0619-035-22). MIT veterinary staff continuously assessed the welfare of the animals prior to, during, and after the experiment.

### Decision letter and Author response

Decision letter https://doi.org/10.7554/eLife.60824.sa1
Author response https://doi.org/10.7554/eLife.60824.sa2

## Additional files

### Supplementary files
• Transparent reporting form

### Data availability

The pre-processed data are available on a public server at: https://github.com/ABastos/Propofol-unconscious-NHP (copy archived at https://archive.softwareheritage.org/swh:1:rev:0f758828628e961707b60c7d7da6849f25c1e708/).

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
