## [Decision Letter]

**Acceptance summary:**

The authors reported that propofol caused a reduction in higher frequencies and increase in lower frequencies following loss of consciousness. The manuscript also displayed that thalamic stimulation greatly reduced slow frequency dynamics. This manuscript suggested that this state of unconsciousness is correlated with the degree of coupling between thalamus and cortex.

**Decision letter after peer review:**

Thank you for submitting your article "Neural effects of propofol-induced unconsciousness and its reversal using thalamic stimulation" for consideration by *eLife*. Your article has been reviewed by 3 peer reviewers, one of whom is a member of our Board of Reviewing Editors, and the evaluation has been overseen by Laura Colgin as the Senior Editor. The reviewers have opted to remain anonymous.

The reviewers have discussed the reviews with one another and the Reviewing Editor has drafted this decision to help you prepare a revised submission.

Reviewer #1:

In the manuscript by Bastos et al., the authors investigated the physiological and neural characteristics of propofol-induced unconscious state in the non-human primates, and the effects of the electrical stimulation in the central thalamus during unconsciousness. The authors described that during unconscious states, slow frequency (0.1-3Hz) and coherence between the cortex and thalamus increase, and spike firing rates decrease stronger in superficial layers 2/3 than deep layers 5/6. High-current thalamic stimulation during loss of consciousness alters physiological signals similar to those in the wake state, and decreases in slow frequency power but increases high frequency (>4Hz) oscillations and spike firing rates in all cortices.

This is a mostly phenomenal, but carefully conducted study.

The authors propose that the synchronization of slow frequency oscillations and entrained low spike activity with Down-states in the intracortical and thalamocortical circuits are important characteristics in unconscious state induced by propofol anesthesia and that the high current thalamic stimulation reverses such characteristics and awakens the non-human primate from unconsciousness. This argument is supported by the analysis of frequency power and coherence of local field potentials (LPFs) and spikes and in the multiple cortical areas and the thalamus throughout the manuscript. However, the author's argument would be stronger if the coherence of frequency power and spike-LFP coupling was reversed in the analysis of the effects of thalamic electrical stimulation.

1. As the authors mentioned in the manuscript, Redinbaugh et al. (2020) has reported the neural correlates of consciousness using propofol anesthesia and the central thalamic electrical stimulation in macaques in the similar manner with this study. It would be necessary to clarify the meaning of this study by comparing the methods and the findings from Redinbaugh et al. (2020) in the introduction and Discussion section more in detail.

2. The anteriorization of alpha frequency oscillation is known as the key feature of propofol-induced general anesthesia in humans. In Figure 4, alpha/beta frequency power in the anterior areas (PFC and 8A) increased during unconscious state compared to those in the posterior regions (PPC and STG). However, it seems confusing that high-current thalamic stimulation also increased alpha/beta frequency power in the 8A and PFC in Figure 9E, which the authors argued to awaken the NHP. The authors need to interpret this finding.

3. The authors argued that thalamic stimulation reverse electrophysiological features of unconsciousness which include slow oscillations, the entrainment of little spiking to Down-state, and decreased coherence in frequencies above 4 Hz. However, to assess the effect of electrical stimulation in the central thalamus the authors compare the change in firing rate and frequency power in each cortical region in Figure 9 and Supple Figure 4. It would be necessary to compare the coherence of frequency power and the coupling between spike and LFP as presented in Figure 4 and Figure 6 in order to support the authors' argument.

Reviewer #2:

The authors show that propofol-induced anesthesia is accompanied by a widespread increase in power in the slow frequency bands (< 4 Hz) in the cortex and coherence in these bands across both the cortex and thalamus. Local cortical spiking is strongly coupled to the Up states of these slow frequency oscillations, and there is a general reduction in firing rate across the cortex. Power in higher-frequency bands (> 40 Hz) was markedly reduced in the cortex, and there was a modest decrease in coherence between some regions in some higher-frequency bands. They then show that thalamic stimulation can modestly reverse some of these electrophysiological signatures of unconscious.

In general, the experimental methodology is solid and the data is consistent with many previous studies examining the neurobiological signatures of anesthesia and unconsciousness. With that said, the manuscript itself does not do much to advance knowledge within the field. Most of the cortical electrophysiological features reported here have been previously reported, and although this study benefits from the ability to analyze spiking in single-units (although see Major Critique 1 as clarification is needed for this study), that local spiking is entrained to Up states necessarily follows from the relationship between the LFP and spiking activity, and is a widely-documented characteristic of slow oscillations. Thalamic activity itself was not characterized (see Major Critique 3), only that there is coherence between it and the cortex in slow frequency bands.

Regarding the effects of thalamic stimulation, these need to be analyzed relative to an awake baseline state in addition to an unconscious baseline (see Critique 2) to claim that it induces an "awake-like cortical state". Finally, the concluding remark that "propofol likely renders unconsciousness by disrupting intracortical and thalamocortical communication through decrease spiking and enhanced slow-frequency power/synchrony", is simply a descriptive characterization of the induced state rather than a mechanistic hypothesis as it is presented.

1. Figure 4 (C,F,I,L). The text describing Figure 4 states "During unconsciousness, spike timing was phase-coupled to the slow frequencies. This appeared as Up and Down states of high vs minimal/no spiking (e.g., Figure 1G) respectively. Spike-triggered averages of the local LFP signal indicated that spikes entrained to the depolarized phases (troughs) of slow frequency oscillations (Figure 4C/F/I/L) in all areas." However, both the figure legend and caption appear to depict that spikes are entrained to slow frequency oscillations to a greater extent during the awake state relative to the unconscious state. Presumably this was a mislabeling of the figures. If not, the author's interpretation of the data is completely unclear and requires explanation.

2. Thalamic stimulation arouses unconscious monkeys and causes a partial reversal of neurophysiological signs of unconsciousness. This section describes that "Stimulation produced an awake-like cortical state", however, all comparisons in the figures are made relative to an unconscious state baseline (with the exception of Figure 9C which depicts the average firing rate at ROC – which is still notably lower than the awake state used as a baseline in previous analyses). All that can be said from this analysis is that stimulation significantly changed activity relative to the unconscious state. If the authors want to make the claim as stated, analysis in Figures 8C-F and 9C-G must be done using relative to an awake state baseline as well. Preferably this would be the pre-drug baseline rather than the ROC, as the ROC is a temporary state which doesn't appear to show any stable phenomena (Figures 1, 2 and 4B).

3. Propofol-induced changed in thalamic activity. Given the focus on the effects of stimulating the thalamus during propofol-induced anesthesia, the authors should at minimum provide a spectrogram of thalamic activity similar to as was done for the cortical regions (see Figure 2), particularly as they speculate that direct inhibition of the thalamus may be partially responsible for the resulting slow-frequency oscillations.

Reviewer #3:

Anesthetic-induced loss-of-consciousness (LoC) remains one, if not the, most pressing unanswered questions in the field of anesthetic mechanisms research. The answer to this question will almost certainly have relevance to our global understanding of phase shifts between consciousness and unconsciousness in more general terms, and is, therefore, of broad interest to a large, and diverse, group of readers. Here, the authors use a non-human primate model (adult macaque monkey) to examine anesthetic (propofol)-induced loss-of-consciousness while recording neuronal spiking activity and local field potentials (LFPs) from a series of chronically implanted 64-channel "Utah" arrays in frontal cortex (8A, PFC), posterior parietal (PPC, 7A/B), and auditory/temporal (superior temporal gyrus, STG) cortex. These initial sessions (ten in monkey 1, 11 in monkey 2) served to establish the neurophysiological properties defining the awake state and the state of unconsciousness. In the second set of experiments, they implanted multiple-contact stimulating electrodes (two in each hemisphere of the same monkeys) in frontal thalamic nuclei (intralaminar nuclei, ILN, and mediodorsal nuclei, MD) to record from and electrically stimulate the thalamus and cortex during propofol-induced unconsciousness (a total of 22 additional sessions, 11 in monkey 1, 11 in monkey 2). In the third set of experiments, in two additional monkeys (a total of eight sessions, two in monkey 3 and six in monkey 4), they performed acute laminar recordings from the same areas using multi-contact arrays positioned approximately perpendicular to cortex.

The experiments were precisely performed and the data rigorously analyzed. The manuscript is clearly written. There are a number of important points that should be addressed:

1. In the Introduction, the paragraph on theories of LoC is by necessity brief, but its brevity presents an incomplete picture of what may be taking place. A useful reference here is the 2019 review by Hemmings et al., which nicely summarizes many of these issues. It is up to the authors if they want to cite the original references therein or rely on the review if space/reference limitations come into play.

2. While the data are clearly informative with regards to propofol-induced LoC (as noted in the Cover Letter), the degree/extent to which the observed patterns of activity and changes in coherence are relevant to other anesthetics is unclear; this is most notably relevant to ketamine and nitrous oxide, both of which are primarily, but not exclusively NMDA receptor antagonists, and at concentrations associated with LoC lead to profound EEG activation. This should be addressed in the Discussion.

3. Results – the cortical recording tracings shown in Figure 1 G (right panel, unconscious) appear to show that the animal is in burst suppression, which occurs at a level of sedation well beyond that required for simple LoC. Where all the subsequent recordings and associated analyses performed at this level of sedation? Was it possible to obtain recordings at a level of sedation that did not produce burst suppression?

4. Confounding variables – As shown in Figure 1F, there is a ~20% decrease in heart rate, which, given the dependency on cardiac output for maintenance of blood pressure, would presumably lead to a corresponding decrease in mean arterial pressure, and therefore cerebral perfusion pressure. Similarly, there is no indication that minute ventilation was controlled via mechanical ventilation. At steady-state levels of infusion in humans propofol produces a 20-50% decrease in minute ventilation (see, for example, Allsop, P., et al. (1988). "Ventilatory effects of a propofol infusion using a method to rapidly achieve steady-state equilibrium." Eur J Anaesthesiol 5(5): 293-303; Goodman, N. W., et al. (1987). "Some ventilatory effects of propofol as sole anaesthetic agent." Br J Anaesth 59(12): 1497-1503). These changes will at the minimum increase PaCO2 by 1 kPa (~ 8 torr), and this too will have an impact on cerebral perfusion. Please explain why these factors do not interfere with the results presented or the interpretation thereof. Simply measuring peripheral capillary oxygen saturation is not the same as knowing what it is in cortical and/or thalamic tissue beds.

---

## [Author Response]

Reviewer #1:[…] The authors propose that the synchronization of slow frequency oscillations and entrained low spike activity with Down-states in the intracortical and thalamocortical circuits are important characteristics in unconscious state induced by propofol anesthesia and that the high current thalamic stimulation reverses such characteristics and awakens the non-human primate from unconsciousness. This argument is supported by the analysis of frequency power and coherence of local field potentials (LPFs) and spikes and in the multiple cortical areas and the thalamus throughout the manuscript. However, the author's argument would be stronger if the coherence of frequency power and spike-LFP coupling was reversed in the analysis of the effects of thalamic electrical stimulation.1. As the authors mentioned in the manuscript, Redinbaugh et al. (2020) has reported the neural correlates of consciousness using propofol anesthesia and the central thalamic electrical stimulation in macaques in the similar manner with this study. It would be necessary to clarify the meaning of this study by comparing the methods and the findings from Redinbaugh et al. (2020) in the introduction and Discussion section more in detail.

We agree with this. Both papers would benefit from this discussion. First, we should point out our papers are in general agreement and that replication is important to science. But there are some noteworthy differences in some methods and results.

1. Redinbaugh et al. used different neural physiological recordings sessions for the awake vs unconscious states.

By contrast, our study used single sessions. We recorded the same neural activity as animals transitioned from awake to unconscious and back again. Thus, we were able to capture neural changes during the transition between states.

2. In the Redinbaugh study, there were several differences between the awake and unconscious recording sessions that could have contributed to some differences in results between our studies.

Redinbaugh administered ketamine two hours before the propofol sessions (but not the awake sessions). Thus, lingering effects of ketamine, and/or its interaction with propofol, could have influenced their results. Plus, in the Redinbaugh study, awake-state recordings were conducted with the animals sitting up in a primate chair while the during the propofol recordings the animals were prone and intubated.

By contrast, we did not use ketamine or any additional drug. We recorded both awake and unconscious states within the same session with no difference between states other than the administration of propofol. Thus, our results can be attributed to propofol alone.

3. The details of the thalamic stimulation differed. Redinbaugh used a single stimulating electrode and found that 50 Hz was most effective at inducing arousal from the unconscious state.

By contrast, we used larger stimulating electrodes in pairs, higher currents, and 180 Hz. We did so because we aimed to match the stimulation parameters of studies showing that thalamic stimulation was effective eliciting responses from minimally conscious humans (Schiff et al., 2007).

Our study and Redinbaugh’s study report similar main results and conclusions. We both report a reduction in higher frequencies and increase in lower frequencies following loss of consciousness from propofol. But they do differ in some details. Redinbaugh report less of a role for slow frequencies (~1 Hz). Likewise, they report that thalamic stimulation increases power/coherence in gamma and alpha (as we do) but found little effect on slow frequencies. We found similar results in gamma and alpha but, by contrast, found that propofol also caused large increases in slow frequency dynamics. Correspondingly, we found that thalamic stimulation greatly reduced slow frequency dynamics.

Any of the differences between studies listed above could explain this somewhat different neural results. It is worth noting that other work from our lab shows that ketamine has a very different effects from propofol, it increases higher-frequency power.

We have added additional text of these points in the Discussion section of our paper. We thank the reviewer for bringing this up, as it is important to contextualize our findings with other recent work.

2. The anteriorization of alpha frequency oscillation is known as the key feature of propofol-induced general anesthesia in humans. In Figure 4, alpha/beta frequency power in the anterior areas (PFC and 8A) increased during unconscious state compared to those in the posterior regions (PPC and STG). However, it seems confusing that high-current thalamic stimulation also increased alpha/beta frequency power in the 8A and PFC in Figure 9E, which the authors argued to awaken the NHP. The authors need to interpret this finding.

We agree that an interpretation of this result would enhance the manuscript. We now provide it in the revised manuscript.

Previous studies have documented that synchronization between cortex and thalamus is prominent in the alpha/beta frequency range (Bastos et al., 2014; Fiebelkorn et al., 2019; Saalmann et al., 2012). Therefore, we propose that by driving the thalamocortical loop via thalamic stimulation, we have enhanced this circuit’s natural resonance frequency. We have added this hypothesis to the revised Discussion.

3. The authors argued that thalamic stimulation reverse electrophysiological features of unconsciousness which include slow oscillations, the entrainment of little spiking to Down-state, and decreased coherence in frequencies above 4 Hz. However, to assess the effect of electrical stimulation in the central thalamus the authors compare the change in firing rate and frequency power in each cortical region in Figure 9 and Supple Figure 4. It would be necessary to compare the coherence of frequency power and the coupling between spike and LFP as presented in Figure 4 and Figure 6 in order to support the authors' argument.

This is an excellent suggestion. It will add value to show how cortico-cortical synchronization changes as a function of thalamic electrical stimulation. Per the reviewer’s suggestion, we have added these analyses.

For all cortico-cortical as well as thalamo-cortical pairs, we analyzed LFP phase synchronization during the high-current thalamic stimulation that induced robust behavioral wakeup. It was compared to LFP synchronization during the prepropofol awake state and to intervals following thalamic stimulation. We used the slow frequency range (0.1-2Hz) where we found strong effects of propofol (and its reversal following thalamic stimulation). Note that by focusing on the reduction of slow frequencies resulting from stimulation, we can rule out any increases in synchrony due to stimulation per se.

We found that 5 of 6 cortico-cortical area pairs and 3 of 4 thalamo-cortical area pairs, thalamic stimulation significantly reduced the magnitude of slow frequency phase synchronization. Amongst these areas there was a 71-77% reduction in thalamocortical slow frequency synchronization and a 29-63% reduction between cortical areas (100% reduction would have been a full return to awake state). We have included this as Figure 9 —figure supplement 3 and 4 as well as accompanying text. We thank the reviewer for this valuable suggestion.

We also analyzed changes in spike-LFP synchrony. We analyzed how spikes in each area changed in their entrainment to the slow oscillation LFP during thalamic stimulation. We found that during thalamic stimulation, slow frequency LFP coupling to spikes was significantly reduced in three out of the 4 areas relative to pre-stimulation unconsciousness. This is now reported in the main text and in Figure 9 —figure supplement 6.

Reviewer #2:[…] In general, the experimental methodology is solid and the data is consistent with many previous studies examining the neurobiological signatures of anesthesia and unconsciousness. With that said, the manuscript itself does not do much to advance knowledge within the field. Most of the cortical electrophysiological features reported here have been previously reported, and although this study benefits from the ability to analyze spiking in single-units (although see Major Critique 1 as clarification is needed for this study), that local spiking is entrained to Up states necessarily follows from the relationship between the LFP and spiking activity, and is a widely-documented characteristic of slow oscillations. Thalamic activity itself was not characterized (see Major Critique 3), only that there is coherence between it and the cortex in slow frequency bands.Regarding the effects of thalamic stimulation, these need to be analyzed relative to an awake baseline state in addition to an unconscious baseline (see Major Critique 2) to claim that it induces an "awake-like cortical state". Finally, the concluding remark that "propofol likely renders unconsciousness by disrupting intracortical and thalamocortical communication through decrease spiking and enhanced slow-frequency power/synchrony", is simply a descriptive characterization of the induced state rather than a mechanistic hypothesis as it is presented.1. Figure 4 (C,F,I,L). The text describing Figure 4 states "During unconsciousness, spike timing was phase-coupled to the slow frequencies. This appeared as Up and Down states of high vs minimal/no spiking (e.g., Figure 1G) respectively. Spike-triggered averages of the local LFP signal indicated that spikes entrained to the depolarized phases (troughs) of slow frequency oscillations (Figure 4C/F/I/L) in all areas." However, both the figure legend and caption appear to depict that spikes are entrained to slow frequency oscillations to a greater extent during the awake state relative to the unconscious state. Presumably this was a mislabeling of the figures. If not, the author's interpretation of the data is completely unclear and requires explanation.

We apologize, this was a mislabeled plot. The slow wave showing strong spike entrainment to negative (depolarized) phases was indeed the unconscious state as the reviewer suspected. This has been corrected and we thank the reviewer for noting it.

2. Thalamic stimulation arouses unconscious monkeys and causes a partial reversal of neurophysiological signs of unconsciousness. This section describes that "Stimulation produced an awake-like cortical state", however, all comparisons in the figures are made relative to an unconscious state baseline (with the exception of Figure 9C which depicts the average firing rate at ROC – which is still notably lower than the awake state used as a baseline in previous analyses). All that can be said from this analysis is that stimulation significantly changed activity relative to the unconscious state. If the authors want to make the claim as stated, analysis in Figures 8C-F and 9C-G must be done using relative to an awake state baseline as well. Preferably this would be the pre-drug baseline rather than the ROC, as the ROC is a temporary state which doesn't appear to show any stable phenomena (Figures 1, 2 and 4B).

We agree. We showed stimulation-induced recovery from the unconscious state but it would also be meaningful to show it relative to the awake state. Per the reviewer’s suggestion, we now do.

We quantified the Return To Wakefulness (RTW) metric as follows: neurophysiological and behavioral measures during pre-drug wakefulness were subtracted from their levels before, during, and after high-current thalamic stimulation. They were then normalized by the level observed during prestimulation baseline. A RTW score of -100% would denote a return to wakefulness levels. A 0% change in the RTW metric would denote no change due to thalamic stimulation. Below, we list the results of thalamic stimulation, also listed in Tables 1 and 2 in the revised manuscript. They indicate a partial return to wakefulness in neural activity and a corresponding partial return in behavioral measures.

Thalamic stimulation had the following effects.

Slow frequencies:

-44 to -89% RTW (return to the awake state) for cortical power, depending on area.

-29 to -63% RTW for cortico-cortical LFP synchrony.

-71 to -77% RTW for thalamo-cortical synchrony.

Beta:

-29% and -44% RTW in power in posterior cortex.

Thalamic stimulation resulted in greater beta power than that seen during the awake state

Gamma:

A -47 to -77% RTW in cortical power.

Spiking (average firing rates): -16% for PFC, -13% for 8A, -33% for PPC, and -26% for STG.

Spike coupling to slow frequency LFP: -24% for PFC, a slight increase for 8A

(stimulation increased spike coupling to SF slightly), -35% for PPC, and -113% for

STG

Behavior: Behavioral measures showed a corresponding partial return to awake state with thalamic stimulation. Evoked eyeblinks: -35% RTW

Blood oxygenation: -69% RTW

Heart rate: slightly higher than awake state by 7-8 beats per minute EMG: No change (but values were highly variable to begin; monkeys bounce between bouts of quiet relaxation and fidgeting when awake).

We have included these analyses as new supplemental figures (Figure 8—figure supplement 2 and Figure 9—figure supplement 1 and 3-8) and we refer to them in the main text. We thank the reviewer for this important suggestion. It has helped us to gain confidence in the interpretation of our central finding, that thalamic stimulation induces a partial return to the pre-drug, awake state.

3. Propofol-induced changed in thalamic activity. Given the focus on the effects of stimulating the thalamus during propofol-induced anesthesia, the authors should at minimum provide a spectrogram of thalamic activity similar to as was done for the cortical regions (see Figure 2), particularly as they speculate that direct inhibition of the thalamus may be partially responsible for the resulting slow-frequency oscillations.

This makes perfect sense and we agree. It is now included

We now provide a spectrogram of the thalamus, per the reviewer’s suggestion. The thalamus showed similar results to the cortex. There was an increase in slow frequency oscillations locked to LOC and a decrease in this same band locked to ROC. In the beta frequency range, thalamic power modulation looked like a mixture between anterior and posterior cortex. There was an initial increase in beta power compared to pre-propofol wakefulness starting at LOC. This was followed by a decrease relative to pre-propofol wakefulness from ~5-14 min post-LOC. Subsequently (from 15 minutes post-LOC to ROC) there was neither an increase nor decrease in beta power relative to pre-propofol wakefulness.

In the gamma frequency range, there were sustained decreases in power between LOC and a few minutes after ROC, similar to all cortical regions. We note that comparing between thalamus and cortex power modulation is slightly complicated by the fact that thalamic electrodes (0.5mm thick stimulating electrodes with low impedance) were different than cortical leads (Utah arrays, very thin silicon contacts with high impedance). We have added these results to Figure 2—figure supplement 2 along with a description of these results in the main text.

Reviewer #3:[…] The experiments were precisely performed and the data rigorously analyzed. The manuscript is clearly written. There are a number of important points that should be addressed:1. In the Introduction, the paragraph on theories of LoC is by necessity brief, but its brevity presents an incomplete picture of what may be taking place. A useful reference here is the 2019 review by Hemmings et al., which nicely summarizes many of these issues. It is up to the authors if they want to cite the original references therein or rely on the review if space/reference limitations come into play.

We have referenced Hemmings review in the revised sections of the introduction.

2. While the data are clearly informative with regards to propofol-induced LoC (as noted in the Cover Letter), the degree/extent to which the observed patterns of activity and changes in coherence are relevant to other anesthetics is unclear; this is most notably relevant to ketamine and nitrous oxide, both of which are primarily, but not exclusively NMDA receptor antagonists, and at concentrations associated with LoC lead to profound EEG activation. This should be addressed in the Discussion.

The reviewer makes a good point. The work we have presented is relevant to the propofol-induced unconsciousness. Propofol’s principal target is GABA-A receptors. Although not tested here, the findings here likely offer insights into the mechanism of unconsciousness induced by other GABAergic anesthetics including the inhaled ethers (sevoflurane, isoflurane desflurane), the barbiturates and etomidate. The current work does not offer insights into the dynamics of NMDA antagonists such as ketamine and nitrous oxide. The agents have noticeably different dynamics at both lower and higher frequencies (Akeju et al., 2016; Pavone et al., 2016; Purdon et al., 2015). We have studied ketamine using our non-human primate model. We have provided a preliminary reports of these results (Garwood et al., 2020). A manuscript on this work is under preparation. We clarify these points in the Discussion.

3. Results – the cortical recording tracings shown in Figure 1 G (right panel, unconscious) appear to show that the animal is in burst suppression, which occurs at a level of sedation well beyond that required for simple LoC. Where all the subsequent recordings and associated analyses performed at this level of sedation? Was it possible to obtain recordings at a level of sedation that did not produce burst suppression?

The analyses were performed at a level of slow-delta oscillations, slow-delta oscillations mixed with burst suppression and burst suppression. The data shown in Figure 1 G (right panel, unconscious) is a mixture of slow-delta oscillations with burst suppressions. We did obtain recordings with a predominance of slow delta oscillations. The mixture of alpha oscillations and slow-delta oscillations that are so prominent in propofol-induced unconsciousness are less apparent in the nonhuman primate. After a brief period of beta oscillations, propofol-induced unconsciousness in the non-human primate transitions to slow-delta oscillations and then into burst suppression. We did trial runs of our sedation protocol in each animal to determine a range of infusions that would produce unconsciousness with the least some amount of burst suppression. The longer the animals were in burst suppression the greater the likelihood that the animal would become apneic and require stimulation or respiratory assistance. For all of the sessions reported here, the animals did not have a period of apnea.

4. Confounding variables – As shown in Figure 1F, there is a ~20% decrease in heart rate, which, given the dependency on cardiac output for maintenance of blood pressure, would presumably lead to a corresponding decrease in mean arterial pressure, and therefore cerebral perfusion pressure. Similarly, there is no indication that minute ventilation was controlled via mechanical ventilation. At steady-state levels of infusion in humans propofol produces a 20-50% decrease in minute ventilation (see, for example, Allsop, P., et al. (1988). "Ventilatory effects of a propofol infusion using a method to rapidly achieve steady-state equilibrium." Eur J Anaesthesiol 5(5): 293-303; Goodman, N. W., et al. (1987). "Some ventilatory effects of propofol as sole anaesthetic agent." Br J Anaesth 59(12): 1497-1503). These changes will at the minimum increase PaCO2 by 1 kPa (~ 8 torr), and this too will have an impact on cerebral perfusion. Please explain why these factors do not interfere with the results presented or the interpretation thereof. Simply measuring peripheral capillary oxygen saturation is not the same as knowing what it is in cortical and/or thalamic tissue beds.

The reviewer makes good point. Because we did not use pressors to maintain blood pressure and mechanical ventilation to oxygenation and ventilation, we readily acknowledge that some component of the observed neurophysiological effects could be due to these metabolic factors. We state this in the revised Discussion.

References:

Akeju O, Song AH, Hamilos AE, Pavone KJ, Flores FJ, Brown EN, Purdon PL. 2016.

Electroencephalogram signatures of ketamine anesthesia-induced unconsciousness.

*Clinical Neurophysiology* 127:2414–2422. doi:10.1016/j.clinph.2016.03.005

Bastos AM, Briggs F, Alitto HJ, Mangun GR, Usrey WM. 2014. Simultaneous recordings from the primary visual cortex and lateral geniculate nucleus reveal rhythmic interactions and a cortical source for γ-band oscillations. *J Neurosci* 34:7639–7644.

doi:10.1523/JNEUROSCI.4216-13.2014

Chemali JJ, Van Dort CJ, Brown EN, Solt K. 2012. Active Emergence from Propofol General Anesthesia Is Induced by Methylphenidate. *Anesthesiology* 116:998–1005.

doi:10.1097/ALN.0b013e3182518bfc

Fiebelkorn IC, Pinsk MA, Kastner S. 2019. The mediodorsal pulvinar coordinates the macaque fronto-parietal network during rhythmic spatial attention. *Nat Commun* 10:1–15. doi:10.1038/s41467-018-08151-4

Garwood IC, Chakravarty S, Donoghue J, Kahali P, Chamadia S, Akeju O, Miller EK, Brown EN. 2020. A hidden Markov model reliably characterizes ketamine-induced spectral dynamics in macaque LFP and human EEG (preprint). Anesthesia.

doi:10.1101/2020.11.12.20221366

Kenny JD, Taylor NE, Brown EN, Solt K. 2015. Dextroamphetamine (but Not Atomoxetine) Induces Reanimation from General Anesthesia: Implications for the Roles of Dopamine and Norepinephrine in Active Emergence. *PLoS ONE* 10:e0131914.

doi:10.1371/journal.pone.0131914

Muindi F, Kenny JD, Taylor NE, Solt K, Wilson MA, Brown EN, Van Dort CJ. 2016. Electrical stimulation of the parabrachial nucleus induces reanimation from isoflurane general anesthesia. *Behav Brain Res* 306:20–25. doi:10.1016/j.bbr.2016.03.021

Pavone KJ, Akeju O, Sampson AL, Ling K, Purdon PL, Brown EN. 2016. Nitrous oxide-induced slow and δ oscillations. *Clinical Neurophysiology* 127:556–564.

doi:10.1016/j.clinph.2015.06.001

Pillay S, Liu X, Baracskay P, Hudetz AG. 2014. Brainstem stimulation increases functional connectivity of basal forebrain-paralimbic network in isoflurane-anesthetized rats. *Brain Connect* 4:523–534. doi:10.1089/brain.2014.0254

Purdon PL, Sampson A, Pavone KJ, Brown EN. 2015. Clinical Electroencephalography for Anesthesiologists. *Anesthesiology* 123:937–960. doi:10.1097/ALN.0000000000000841

Saalmann YB, Pinsk MA, Wang L, Li X, Kastner S. 2012. The Pulvinar Regulates Information Transmission Between Cortical Areas Based on Attention Demands. *Science* 337:753– 756. doi:10.1126/science.1223082

Schiff ND, Giacino JT, Kalmar K, Victor JD, Baker K, Gerber M, Fritz B, Eisenberg B, Biondi T, O’Connor J, Kobylarz EJ, Farris S, Machado A, McCagg C, Plum F, Fins JJ, Rezai AR. 2007. Behavioural improvements with thalamic stimulation after severe traumatic brain injury. *Nature* 448:600–603. doi:10.1038/nature06041

Solt K, Cotten JF, Cimenser A, Wong KFK, Chemali JJ, Brown EN. 2011. Methylphenidate Actively Induces Emergence from General Anesthesia. *Anesthesiology* 115:791–803.

doi:10.1097/ALN.0b013e31822e92e5

Solt K, Van Dort CJ, Chemali JJ, Taylor NE, Kenny JD, Brown EN. 2014. Electrical stimulation of the ventral tegmental area induces reanimation from general anesthesia.

*Anesthesiology* 121:311–319. doi:10.1097/ALN.0000000000000117

Taylor NE, Chemali JJ, Brown EN, Solt K. 2013. Activation of D1 Dopamine Receptors Induces Emergence from Isoflurane General Anesthesia. *Anesthesiology* 118:30–39.

doi:10.1097/ALN.0b013e318278c896

Taylor NE, Van Dort CJ, Kenny JD, Pei J, Guidera JA, Vlasov KY, Lee JT, Boyden ES, Brown EN, Solt K. 2016. Optogenetic activation of dopamine neurons in the ventral tegmental area induces reanimation from general anesthesia. *Proc Natl Acad Sci USA* 113:12826–12831. doi:10.1073/pnas.1614340113